# The NAD+ precursor NMN activates dSarm to trigger axon degeneration in *Drosophila*

**Arnau Llobet Rosell[1], Maria Paglione[1], Jonathan Gilley[2], Magdalena Kocia[1], Giulia Perillo[3], Massimiliano Gasparrini[4], Lucia Cialabrini[4], Nadia Raffaelli[4], Carlo Angeletti[5], Giuseppe Orsomando[5], Pei-Hsuan Wu[3], Michael P Coleman[2], Andrea Loreto[2†], Lukas Jakob Neukomm[1]\***

[1]Department of Fundamental Neurosciences, University of Lausanne, Lausanne, Switzerland; [2]John van Geest Centre for Brain Repair, Department of Clinical Neurosciences, University of Cambridge, Cambridge, United Kingdom; [3]Department of Genetic Medicine and Development, University of Geneva, Geneva, Switzerland; [4]Department of Agricultural, Food and Environmental Sciences, Polytechnic University of Marche, Ancona, Italy; [5]Department of Clinical Sciences, Section of Biochemistry, Polytechnic University of Marche, Ancona, Italy

**\*For correspondence:**
lukas.neukomm@unil.ch

**Present address:** [†]Department of Physiology, Anatomy and Genetics, Kavli Institute for NanoScience Discovery, Oxford Parkinson's Disease Centre, University of Oxford, Oxford, United Kingdom

**Competing interest:** The authors declare that no competing interests exist.

**Abstract** Axon degeneration contributes to the disruption of neuronal circuit function in diseased and injured nervous systems. Severed axons degenerate following the activation of an evolutionarily conserved signaling pathway, which culminates in the activation of SARM1 in mammals to execute the pathological depletion of the metabolite NAD+. SARM1 NADase activity is activated by the NAD+ precursor nicotinamide mononucleotide (NMN). In mammals, keeping NMN levels low potently preserves axons after injury. However, it remains unclear whether NMN is also a key mediator of axon degeneration and dSarm activation in flies. Here, we demonstrate that lowering NMN levels in *Drosophila* through the expression of a newly generated prokaryotic NMN-Deamidase (NMN-D) preserves severed axons for months and keeps them circuit-integrated for weeks. NMN-D alters the NAD+ metabolic flux by lowering NMN, while NAD+ remains unchanged in vivo. Increased NMN synthesis by the expression of mouse nicotinamide phosphoribosyltransferase (mNAMPT) leads to faster axon degeneration after injury. We also show that NMN-induced activation of dSarm mediates axon degeneration in vivo. Finally, NMN-D delays neurodegeneration caused by loss of the sole NMN-consuming and NAD+-synthesizing enzyme dNmnat. Our results reveal a critical role for NMN in neurodegeneration in the fly, which extends beyond axonal injury. The potent neuroprotection by reducing NMN levels is similar to the interference with other essential mediators of axon degeneration in *Drosophila*.

## Editor's evaluation

The paper by Llobet Rosell et al. and the tools they develop will be valuable to the neurodegeneration/axon injury field. The authors systematically analyze the signaling role of NMN, a NAD precursor metabolite in activating the axon degeneration trigger dSARM, in *Drosophila*. They show the extent to which the NMN/NAD+ ratio drives pathological axon degeneration and demonstrate convincingly, that reducing NMN levels is strongly protective in several in vivo injury paradigms.

## Introduction

The elimination of large portions of axons is a widespread event in the developing nervous system (*Neukomm and Freeman, 2014*; *Riccomagno and Kolodkin, 2015*). Axon degeneration is also an early hallmark of nervous system injury and a common feature of neurodegenerative diseases (*Coleman and Höke, 2020*; *Mariano et al., 2018*; *Merlini et al., 2022*). Understanding the underlying molecular mechanisms may facilitate the development of treatments to block axon loss in acute or chronic neurological conditions.

Wallerian degeneration is a well-established, evolutionarily conserved, and simple system to study how injured axons execute their self-destruction (*Llobet Rosell and Neukomm, 2019*; *Waller, 1850*). Upon axonal injury (axotomy), distal axons separated from their soma degenerate within a day. Axotomy activates a signaling pathway (programmed axon degeneration, or axon death) that actively executes the self-destruction of severed axons. Induced signaling culminates in the activation of sterile alpha and TIR motif-containing protein 1 (dSarm and SARM1 in flies and mice, respectively) (*Gerdts et al., 2013*; *Osterloh et al., 2012*). As NADase, once activated, dSarm/SARM1 executes the pathological depletion of nicotinamide adenine dinucleotide ($NAD^+$) in severed axons, culminating in catastrophic fragmentation (*Essuman et al., 2017*; *Figley et al., 2021*; *Gerdts et al., 2015*). Initially thought to be activated only after injury, evidence accumulated over recent years that axon death signaling is also activated in many non-injury neurological disorders (*Figley and DiAntonio, 2020*; *Hopkins et al., 2021*).

In mammals, SARM1 activation is tightly controlled by metabolites in the $NAD^+$ biosynthetic pathway. The labile enzyme nicotinamide mononucleotide adenylyltransferase 2 (NMNAT2) is constantly transported into the axon, where it is degraded by the E3 ubiquitin ligase PAM-Highwire-Rpm-1 (PHR1) and mitogen-activated protein kinase (MAPK) signaling (*Babetto et al., 2013*; *Gilley and Coleman, 2010*; *Walker et al., 2017*). This steady state results in sufficient NMNAT2 that consumes nicotinamide mononucleotide (NMN) to synthesize $NAD^+$. Upon axonal injury, axonal transport halts. Subsequently, NMNAT2 is rapidly degraded. It leads to a temporal rise of axonal NMN and a halt in $NAD^+$ biosynthesis (*Di Stefano et al., 2017*; *Di Stefano et al., 2015*; *Loreto et al., 2021*; *Loreto et al., 2015*). NMN and $NAD^+$ compete by binding to an allosteric pocket in the SARM1 armadillo-repeat (ARM) domain, which is crucial for SARM1 activation. While a rise in NMN activates SARM1 by inducing its conformational change (*Bratkowski et al., 2020*; *Figley et al., 2021*; *Zhao et al., 2019*), $NAD^+$ prevents this activation by competing for the same pocket in the ARM domain (*Jiang et al., 2020*). This competitive binding occurs at physiologically relevant levels of NMN and $NAD^+$ (*Angeletti et al., 2022*).

Previous studies have shown that expression of the prokaryotic enzyme PncC–also known as NMN-Deamidase (NMNd)–converts NMN to nicotinic acid mononucleotide (NaMN) (*Galeazzi et al., 2011*), prevents SARM1 activation and preserves severed axons in mammals and zebrafish: for instance, axons with NMNd remain preserved up to 96 hr in murine neuronal cultures (*Di Stefano et al., 2015*; *Loreto et al., 2015*; *Sasaki et al., 2016*), 24 hr in zebrafish and 3 weeks in mice (*Di Stefano et al., 2017*). It remains currently unclear whether NMNd expression levels determine the extent of preservation.

Much of this mechanism of axon death signaling is conserved in *Drosophila* (*Llobet Rosell and Neukomm, 2019*). However, flies harbor some notable differences. A single *dnmnat* gene provides a cytoplasmic and nuclear splice protein variant; consequently, *dnmnat* disruption results in cellular dNmnat loss (*Ruan et al., 2015*). dNmnat turnover is regulated solely by the E3 ubiquitin ligase Highwire (Hiw) (*Xiong et al., 2012*) but not by MAPK signaling (*Neukomm et al., 2017*). Furthermore, the BTB/Back domain-containing Axundead (Axed) executes catastrophic fragmentation downstream of $NAD^+$ depletion, while the mammalian paralog(s) remain to be identified (*Neukomm et al., 2017*).

The role of NMN in axon degeneration in *Drosophila* is controversial. Flies lack nicotinamide phosphoribosyltransferase (NAMPT) (*Gossmann et al., 2012*). NMN might, therefore, be a dispensable intermediate in the $NAD^+$ biosynthetic pathway, thus playing a minor role as a mediator of axon degeneration (*Gerdts et al., 2016*). The Gal4/UAS-mediated NMNd expression in *Drosophila* neurons preserves severed axons for 3–5 days after injury (*Hsu et al., 2021*). It contrasts with the phenotype of other axon death signaling mediators, such as loss-of-function mutations in *hiw*, *dsarm*, and *axed*, as well as over-expression of *dnmnat* (*dnmnat$^{OE}$*), all of which harbor severed axons that remain preserved for weeks to months (*Fang et al., 2012*; *Neukomm et al., 2017*; *Neukomm et al., 2014*;

*Osterloh et al., 2012*). Therefore, the role of NMN as an axon death mediator in *Drosophila* remains to be formally determined.

Here, we report that NMN is an essential mediator of injury-induced axon degeneration in *Drosophila*. Genetic modifications resulting in low NMN levels protect severed axons for the lifespan of the fly, while the addition of an extra NMN synthesizing activity forces axons to undergo faster degeneration after injury. NMN induces the dSarm NADase activity, demonstrating its role as a crucial activator in vivo.

## Results

### Robust expression of prokaryotic NMN-Deamidase in *Drosophila*

Mutations in *hiw*, *dsarm*, or *axed* attenuate axon death signaling resulting in morphological preservation of severed axons for approximately 50 days, the average lifespan of *Drosophila* (*Neukomm et al., 2017*; *Neukomm et al., 2014*; *Osterloh et al., 2012*). In contrast, neuronal expression of prokaryotic NMN-Deamidase (NMNd) to consume NMN results in less than 10% of severed axons being preserved at 7 days post axotomy (dpa) (*Hsu et al., 2021*). We performed an established wing injury assay to confirm this observation (*Paglione et al., 2020*). Briefly, a subset of GFP-labeled sensory neurons (e.g. *dpr1–Gal4* MARCM clones) expressing NMNd (*Hsu et al., 2021*) or GFP were subjected to axotomy in 5- to 7-day-old flies with one wing being partially injured and the other serving as an uninjured control (*Figure 1—figure supplement 1A*). At 7 dpa, we quantified uninjured control axons, axonal debris and severed intact axons, respectively (*Figure 1—figure supplement 1B*, left), and calculated the percentage of protected severed axons (*Figure 1—figure supplement 1B*, right). We observed a 40% preservation of severed axons with NMNd (*Figure 1—figure supplement 1*, genotypes in *Source data 1*). The modest increase of preservation in our hands is probably due to higher NMNd levels caused by higher Gal4 levels in *dpr1* than *elaV* (*Hsu et al., 2021*). However, the expression of NMNd fails to attenuate axon death signaling to the extent of axon death mutants, suggesting that NMN does not play an essential role in activating axon death supported by the absence of the NMN-synthesizing enzyme Nampt in flies (*Gossmann et al., 2012*). Alternatviely, NMNd expression and the resulting NMN consumption is simply not sufficient for potent attenuation of axon death signaling and preservation of severed axons.

Based on the above observations, we decided to generate an N-terminal GFP-tagged NMN-Deamidase (GFP::NMN-D) to increase protein stability (*Rücker et al., 2001*). Plasmids with GFP-tagged wild-type and enzymatically dead versions of NMN-Deamidase were generated (*Di Stefano et al., 2017*; *Di Stefano et al., 2015*), under the control of the UAS regulatory sequence (*UAS–GFP::NMN-D*, and *UAS–GFP::NMN-D*[dead], respectively). We found that wild-type and enzymatically dead NMN-D enzymes are equally expressed in S2 cells, as detected by our newly generated anti-PncC/NMNd/NMN-D antibodies (Materials and methods, *Figure 1—figure supplement 2*). Notably, we observed two immunoreactivities per lane, with the lower band being a potential degradation product.

The similar expression of the NMN-D enzymes prompted us to use the plasmids to generate transgenic flies by targeted insertion (*attP40* landing site). To compare in vivo expression levels, NMN-D, NMN-D[dead], and NMNd were expressed with pan-cellular *actin–Gal4*. We found that NMN-D and NMN-D[dead] immunoreactivities were significantly stronger than NMNd (*Figure 1—figure supplement 2B,C*). In addition, GFP immunoreactivity was also detected in NMN-D and NMN-D[dead], confirming the robust expression of the newly generated tagged proteins (*Figure 1A*). These results show that our newly generated GFP-tagged NMN-D variants are substantially stronger expressed than NMNd in vivo.

### Neuronal NMN-D expression blocks injury-induced axon degeneration for the lifespan of *Drosophila*

The lower-expressed NMNd resulted in a 40% preservation in our wing injury assay. We repeated the injury assay to assess the preservation of our newly generated and higher-expressed NMN-D variants. While severed axons with GFP or NMN-D[dead] degenerated, axons with NMN-D remained fully preserved at 7 dpa (*Figure 1B and C*, *Figure 1—figure supplement 3*). This contrasts with the weaker preservation of axons with lower NMNd expression (*Figure 1—figure supplement 1*). Similarly,

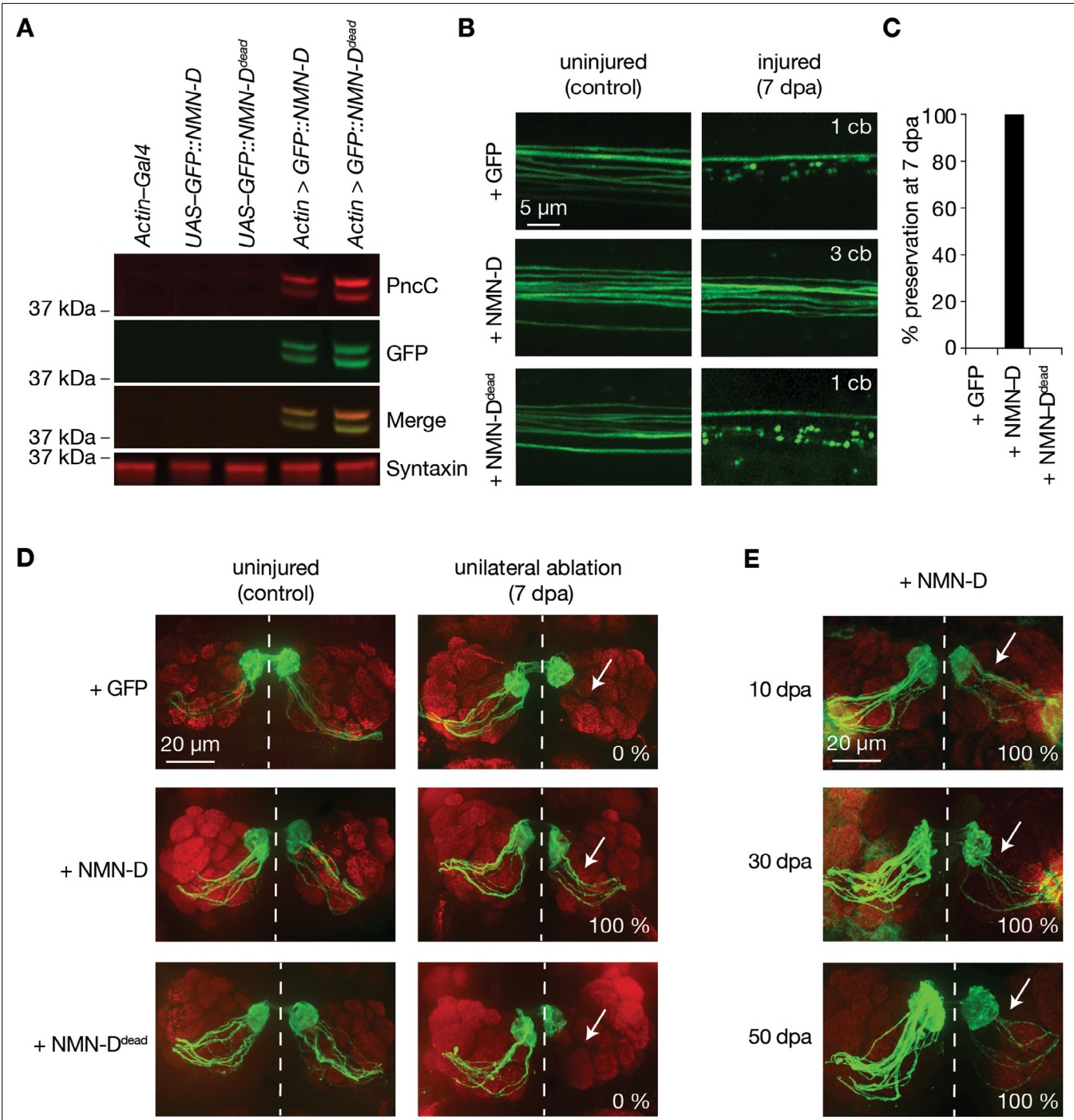

**Figure 1.** Neuronal expression of prokaryotic NMN-D preserves the morphology of severed axons for the lifespan in *Drosophila*. (**A**) Equal expression of wild-type and enzymatically dead NMN-D enzymes, respectively. Western blots with anti-PncC and anti-GFP immunoreactivities (red and green, respectively). (**B**) Low NMN results in severed wing sensory neuron axons that remain morphologically preserved at 7 dpa. Examples of control and 7 dpa. (**C**) Axon death quantification. % preservation of injured axons at 7 dpa, average ± standard error of the mean (n=15 wings). (**D**) Low NMN results in severed axons of olfactory receptor neurons that remain morphologically preserved at 7 dpa. Examples of control and 7 dpa (arrows, site of unilateral ablation). Lower right, % of brains with severed preserved axon fibers. (**E**) Low NMN results in severed axons that remain morphologically preserved for 50 days. Representative pictures of 10, 30, and 50 dpa, from a total of 10 brains imaged for each condition (arrows, site of unilateral ablation). Lower right, % of brains with severed preserved axon fibers.

The online version of this article includes the following source data and figure supplement(s) for figure 1:

**Source data 1.** Raw data of Western blot and quantification.

*Figure 1 continued on next page*

*Figure 1 continued*

**Figure supplement 1.** Partial preservation of severed axons at 7 dpa by previously published NMN-Deamidase (NMNd).

**Figure supplement 1—source data 1.** Raw data of quantification.

**Figure supplement 2.** Increased NMN-D detected by anti-PncC antibodies.

**Figure supplement 2—source data 1.** Raw data of Western blot and quantification.

**Figure supplement 3.** Quantification of axonal phenotypes.

**Figure supplement 3—source data 1.** Raw data of quantification.

---

strong preservation was seen in cholinergic olfactory receptor neurons (ORNs), where severed axons with NMN-D remained preserved at 7 dpa (*Figure 1D*). We extended the ORN injury assay and found preservation at 10, 30, and 50 dpa (*Figure 1E*). While quantifying the precise number of axons is technically not feasible, severed preserved axons were observed in all 10, 30, and 50 dpa brains, albeit fewer at later time points (*MacDonald et al., 2006*). Thus, high levels of NMN-D confer robust protection of severed axons for multiple neuron types for the entire lifespan of *Drosophila*.

## NMN-D alters the NAD+ metabolic flux to lower NMN in fly heads

Before measuring the effect of NMN-D on the NAD+ metabolic flux, we measured the activities of the various NAD+ biosynthetic enzymes in fly heads (*Figure 2A*, *Figure 2—figure supplement 1A*, *Amici et al., 2017*; *Zamporlini et al., 2014*). We confirmed that NAD+ can be synthesized from nicotinamide (NAM), nicotinamide riboside (NR), and quinolinate. However, the *Drosophila* Qaprt homolog that catalyzes the conversion from quinolinate to nicotinic acid mononucleotide (NaMN) remains to be identified (*Katsyuba et al., 2018*). We also confirmed the absence of NAMPT activity (*Figure 2—figure supplement 1A*, *Gossmann et al., 2012*). In addition, we confirmed the expression of genes involved in NAD+ synthesis and axon death signaling that are involved in NAD+ metabolism by measuring respective mRNA abundance in fly heads by qRT-PCR (*Figure 2—figure supplement 1B*). We note that the expression and activity of NAD+ metabolic enzymes can be readily detected fly heads.

Next, we wanted to know whether the sole expression of NMN-D can change the NAD+ metabolic flux in vivo (*Figure 2A*). We compared levels of metabolites in heads by LC-MS/MS among samples that expressed NMN-D and NMN-D$^{dead}$ (*Figure 2B*, *van der Velpen et al., 2021*). Consistent with robust NMN-D activity, NMN levels were sixfold lower and NaMN 12-fold higher. We also found significantly higher NaAD and NaR levels. Importantly, all other metabolites remained unchanged, including NAD+ (*Figure 2B*).

Prompted by such a significant change in the NAD+ metabolic flux, we wondered whether the change could alter the expression of genes involved in NAD+ metabolism or axon death signaling. However, besides the expected significant increase of the Gal4-mediated expression of NMN-D and NMN-D$^{dead}$, we did not observe any notable changes (*Figure 2—figure supplement 2*). Our observations demonstrate that the expression of NMN-D alone is sufficient to change the NAD+ metabolic flux, thereby significantly lowering NMN levels without affecting NAD+ in *Drosophila* heads; they serve as an excellent tissue for metabolic analyses.

## Low axonal NMN preserves synaptic connectivity for weeks after injury

Mutations that attenuate axon death signaling robustly suppress the morphological degeneration after axotomy. They also preserve synaptic connectivity. We have previously demonstrated that synaptic connectivity of severed axons with attenuated axon death remains preserved for at least 14 days using an established optogenetic assay (*Neukomm et al., 2017*; *Paglione et al., 2020*). Briefly, mechanosensory chordotonal neurons in the Johnston's organ (JO), whose cell bodies are in the second segment of adult antennae, are required and sufficient for antennal grooming (*Hampel et al., 2015*; *Seeds et al., 2014*). The JO-specific expression of CsChrimson combined with a red-light stimulus can specifically and robustly induce antennal grooming.

We used this assay to test individual flies before bilateral antennal ablation (ctl) and at 7 dpa (*Figure 3*). Flies expressing GFP in JO neurons failed to elicit antennal grooming following red-light exposure at 7 dpa (*Figure 3*, *Video 1*). In contrast, flies with JO-specific NMN-D expression continued to elicit antennal grooming at 7 dpa (*Figure 3*, *Video 2*). Remarkably, the evoked grooming behavior

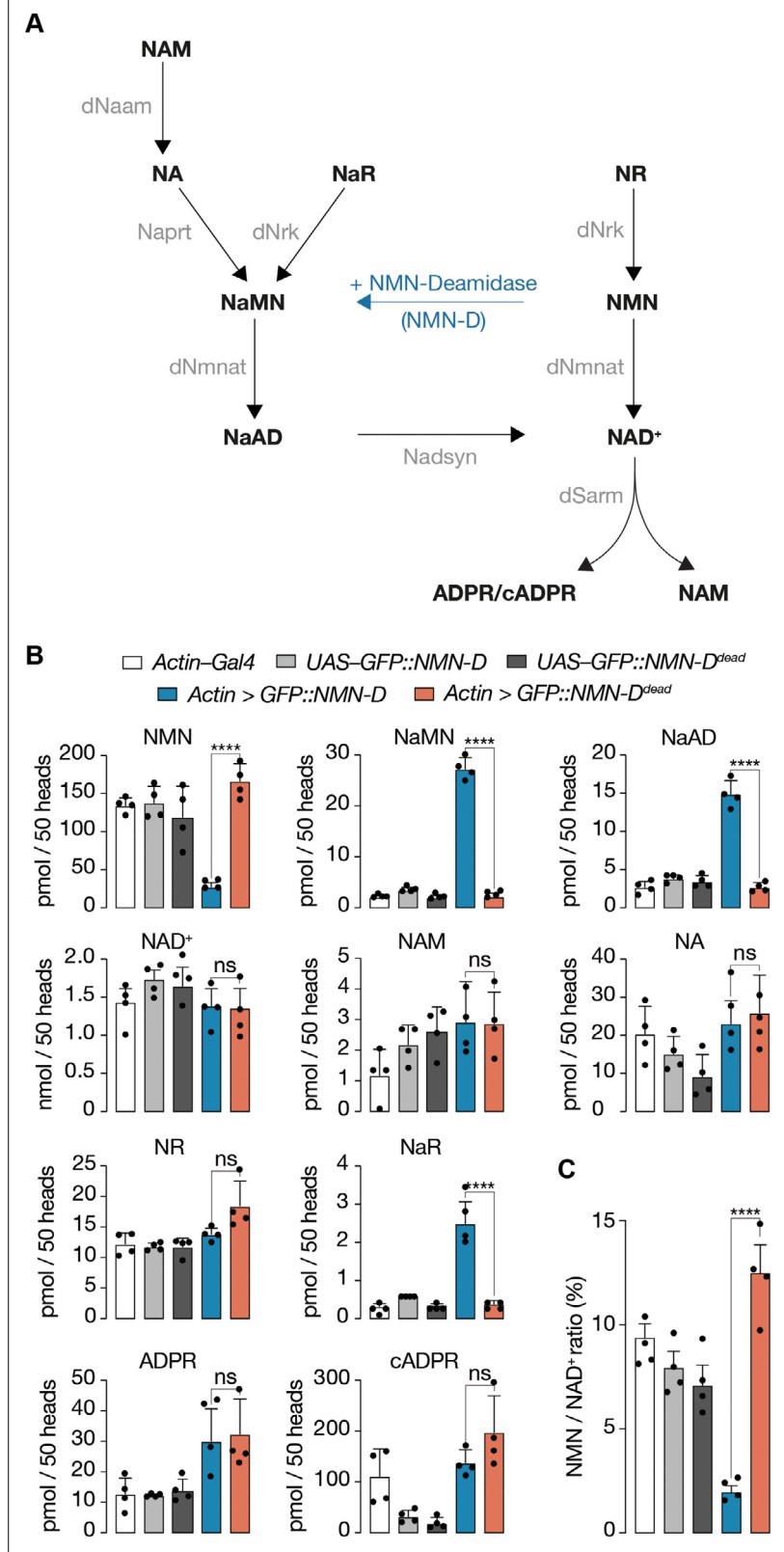

**Figure 2.** Pan-cellular NMN-D expression alters the flux of NAD$^+$ metabolites to lower NMN in heads of *Drosophila*. (**A**) *Drosophila* NAD$^+$ metabolic pathway. Black, metabolites; grey, enzymes; blue, prokaryotic NMN-D, respectively. (**B**) The expression of NMN-D results in lower NMN and higher NaMN, NaAD, and NaR levels, respectively. (**C**) % NMN / NAD$^+$ ratio. Extracted metabolites from 50 heads, mean ± standard deviation (n=4).

*Figure 2 continued on next page*

*Figure 2 continued*

Dots, individual measurements. One-way ANOVA with Tukey's multiple comparisons test, ****=p < 0.0001, ns = p > 0.05.

The online version of this article includes the following source data and figure supplement(s) for figure 2:

**Source data 1.** Raw data of measured metabolites and quantification.

**Figure supplement 1.** Enzymatic activity and expression levels of NAD$^+$ synthesis and axon death genes.

**Figure supplement 1—source data 1.** Raw data of enzymatic activity and qRT-PCR.

**Figure supplement 2.** Lower NMN does not alter mRNA abundance of axon death or NAD$^+$ synthesis genes in *Drosophila* heads.

**Figure supplement 2—source data 1.** Raw data of qRT-PCR.

remained equally robust at 14 dpa (*Figure 3—figure supplement 1*). Preservation of synaptic connectivity depended on low NMN levels, as flies expressing NMN-D$^{dead}$ in JO neurons failed to elicit antennal grooming upon red-light exposure at 7 dpa (*Figure 3*, *Video 3*). Our findings demonstrate that lowering NMN potently attenuates axon death signaling, which is sufficient to preserve synaptic connectivity of severed axons and synapses for weeks after injury.

## mNAMPT-expressing axons degenerate faster after injury

Two enzymatic reactions synthesize NMN in mammals: NAMPT-mediated NAM and NRK1/2-mediated NR consumption. *Drosophila* lacks NAMPT activity, and NMN synthesis relies solely on Nrk-mediated NR consumption (*Figure 2—figure supplement 1A*). We hypothesized that mouse NAMPT (mNAMPT) expression could increase NMN synthesis and, therefore, lead to faster injury-induced axon degeneration in vivo (*Figure 4A*). We generated transgenic flies harboring mNAMPT under the control of UAS by targeted insertion (*attP40*). Western blots revealed proper expression of mNAMPT in fly heads (*Figure 4B*).

We then tested the effect of mNAMPT on the NAD$^+$ metabolic flux in vivo. Surprisingly, NAM, NMN, and NAD$^+$ levels remained unchanged under physiological conditions (*Figure 4C*). However, we noticed threefold higher NR and a moderate but significant elevation of ADPR and cADPR levels upon mNAMPT overexpression (*Figure 4C*). We also asked whether mNAMPT impacts on NAD$^+$ homeostasis thereby altering the expression of axon death or NAD$^+$ synthesis genes. Besides the expected significant increase in the Gal4-mediated expression of *mNAMPT*, we did not observe any notable changes at the mRNA level (*Figure 4—figure supplement 1*).

Although mNAMPT expression failed to elevate NMN under physiological conditions, we hypothesized that mNAMPT could boost NMN levels after injury because of the following observations in mammals: NMNAT2 is labile and rapidly degraded in severed axons (*Gilley and Coleman, 2010*), while NAMPT persists much longer (*Di Stefano et al., 2015*). We speculated that in flies, in severed axons, dNmnat declines similarly, but not mNAMPT. Consequently, NMN accumulates. Strikingly, in our wing injury assay, while axons with GFP showed signs of degeneration starting from 6 hr post axotomy (hpa), mNAMPT expression resulted in significantly faster axon degeneration with signs of degeneration at 4 hpa (*Figure 4D and E*). This accelerated degeneration is likely linked to increased NMN production, but other mechanisms cannot be excluded as there is no increase in NMN under physiological conditions.

We next asked whether the faster degeneration of mNAMPT-expressing severed axons requires axon death genes. While mutations in *dsarm* and *hiw* completely blocked the degeneration of severed axons expressing mNAMPT, *axed* showed a partial preservation of 60% at 12 hr after injury (*Figure 4F*). Importantly, *axed* mutants, in the absence of mNAMPT expression, showed a similar preservation within the first 12 hr (*Figure 4—figure supplement 2*). This preservation remained unchanged at 7 dpa, suggesting that mNAMPT expression does not change the preservation provided by *axed*, *dsarm*, and *hiw* (*Figure 4—figure supplement 2*, *Figure 4G*). Our observations support that elevated NMN levels require axon death signaling to initiate the degeneration of severed axons.

Overall, our data suggest that NMN accumulation after injury triggers axon degeneration in *Drosophila* through the axon death pathway. To the best of our knowledge, we provide the first direct in vivo demonstration that an additional source of NMN synthesis–by the expression of mNAMPT–accelerates injury-induced axon degeneration.

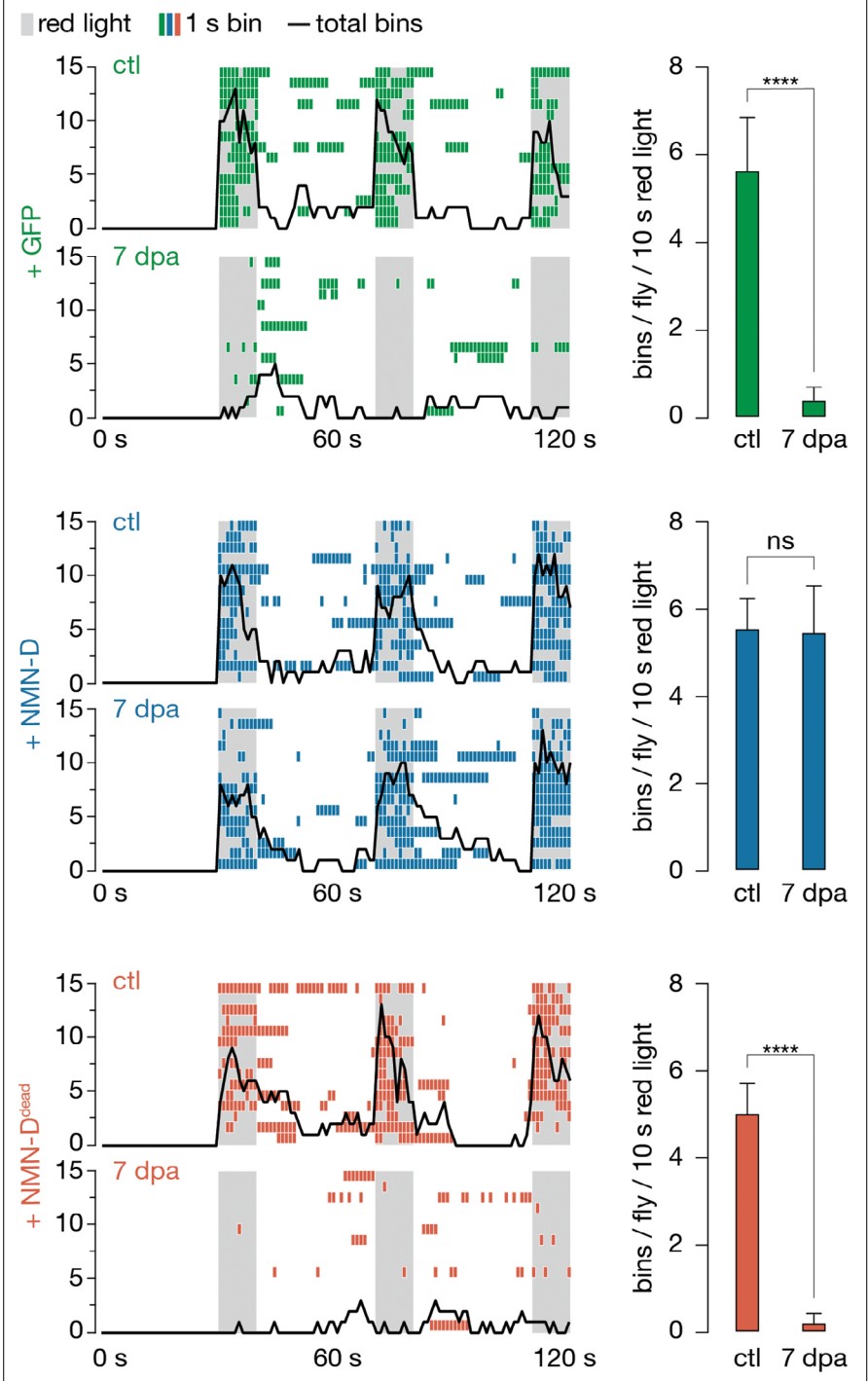

**Figure 3.** Low neuronal NMN preserves synaptic connectivity at 7 dpa. Antennal grooming induced by red light. Left: ethograms of uninjured control (ctl) and 7 dpa flies. Gray bars, 10 s red light; colored boxes, bins; black line, sum of bins (n=15 flies). Right: average bins per fly during 10 s red-light exposure (n=15 flies). Two-tailed t-student test, ****=p < 0.0001, ns = p > 0.05.

The online version of this article includes the following source data and figure supplement(s) for figure 3:

**Source data 1.** Raw data of grooming.

**Figure supplement 1.** Low neuronal NMN preserves synaptic connectivity for weeks after injury.

**Figure supplement 1—source data 1.** Raw data of grooming.

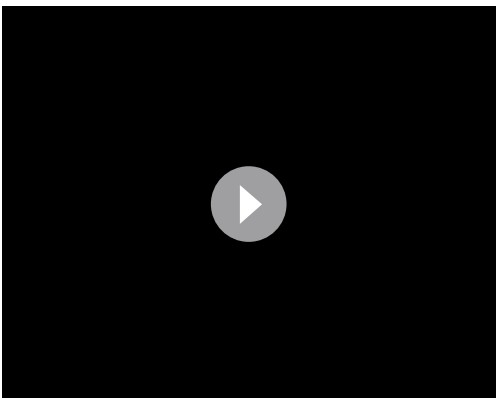

**Video 1.** Examples of red light-stimulated wild-type flies expressing CsChrimson and GFP in JO neurons, uninjured and at 7 dpa. Chamber diameter, 2 cm.
https://elifesciences.org/articles/80245/figures#video1

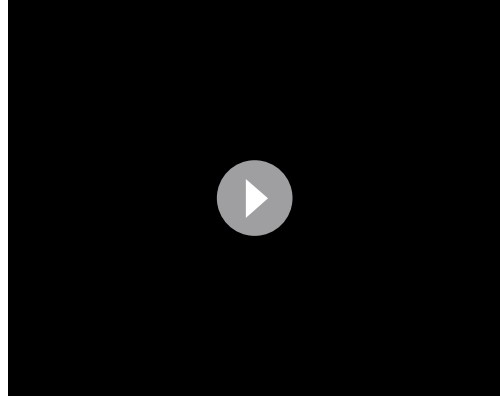

**Video 2.** Examples of red light-stimulated wild-type flies expressing CsChrimson and GFP::NMN-D in JO neurons, uninjured and at 7 dpa. Chamber diameter, 2 cm.
https://elifesciences.org/articles/80245/figures#video2

## NMN activation of dSarm NADase is required for axon degeneration in vivo

We have shown that low NMN attenuates injury-induced axon degeneration, while a more rapid accumulation of NMN, due to expression of mNAMPT, results in faster injury-induced axon degeneration. A cell-permeable form of NMN (CZ-48) binds to and activates SARM1 by changing its conformation (*Zhao et al., 2019*). Crystal structures of the ARM domain of dSarm (dSarm^ARM), as well as the full-length human SARM1 (hSARM1), support the observation that NMN acts as a ligand for dSarm^ARM (*Figley et al., 2021*; *Gu et al., 2021*). NMN binding to the ARM domain requires a critical residue, lysine 193 (K193), to induce a conformational change in the ARM domain of dSarm/SARM1. Consistent with this, a mutation of the lysine residue (e.g. K193R) results in a dominant-negative injury-induced axon degeneration phenotype in murine cell cultures (*Bratkowski et al., 2020*; *Figley et al., 2021*; *Geisler et al., 2019*; *Loreto et al., 2021*; *Zhao et al., 2019*).

To confirm whether NMN activates dSarm in vitro and in vivo, we generated *dsarm* constructs encoding wild-type and the human K193R-equivalent K450R mutation. Crucially, an isoform we used previously, dSarm(D), fails to fully rescue the *dsarm^896* defective axon death phenotype (*Figure 5—figure supplement 1A, B*, *Neukomm et al., 2017*). Therefore, among the eight distinct *dsarm* transcripts, which all contain the ARM, SAM, and TIR domains, we chose the shortest coding isoform, *dsarm(E)* (*Figure 6—figure supplement 1A*). We generated untagged and C-terminal FLAG-tagged dSarm(E), with and without K450R, under the control of UAS and confirmed the FLAG-tagged proteins are expressed at similar levels in S2 cells (*Figure 5A*). We also directly tested the immunopurified FLAG-tagged proteins for constitutive and NMN-inducible NADase activity (*Figure 5B*). While wild-type and K450R dSarm(E) had similar constitutive activities in the presence of 25 μM NAD^+ alone, we found that only wild-type NADase activity was induced further with the addition of 50 μM NMN. At the same time, K450R remained essentially unchanged (*Figure 5C*). This confirmed the critical role of K450 in NMN-dependent activation of the dSarm NADase, equivalent to the role of K193 in hSARM1 (*Figley et al., 2021*; *Loreto et al., 2021*).

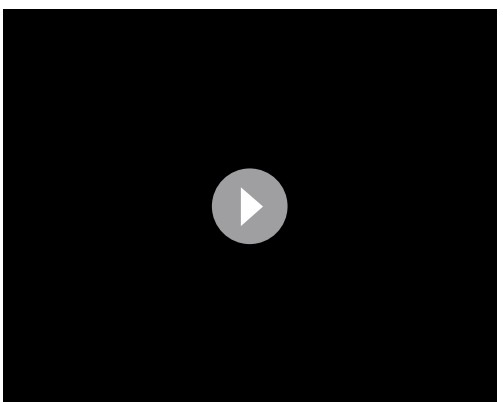

**Video 3.** Examples of red light-stimulated wild-type flies expressing CsChrimson and GFP::NMN-D^dead in JO neurons, uninjured and at 7 dpa. Chamber diameter, 2 cm.
https://elifesciences.org/articles/80245/figures#video3

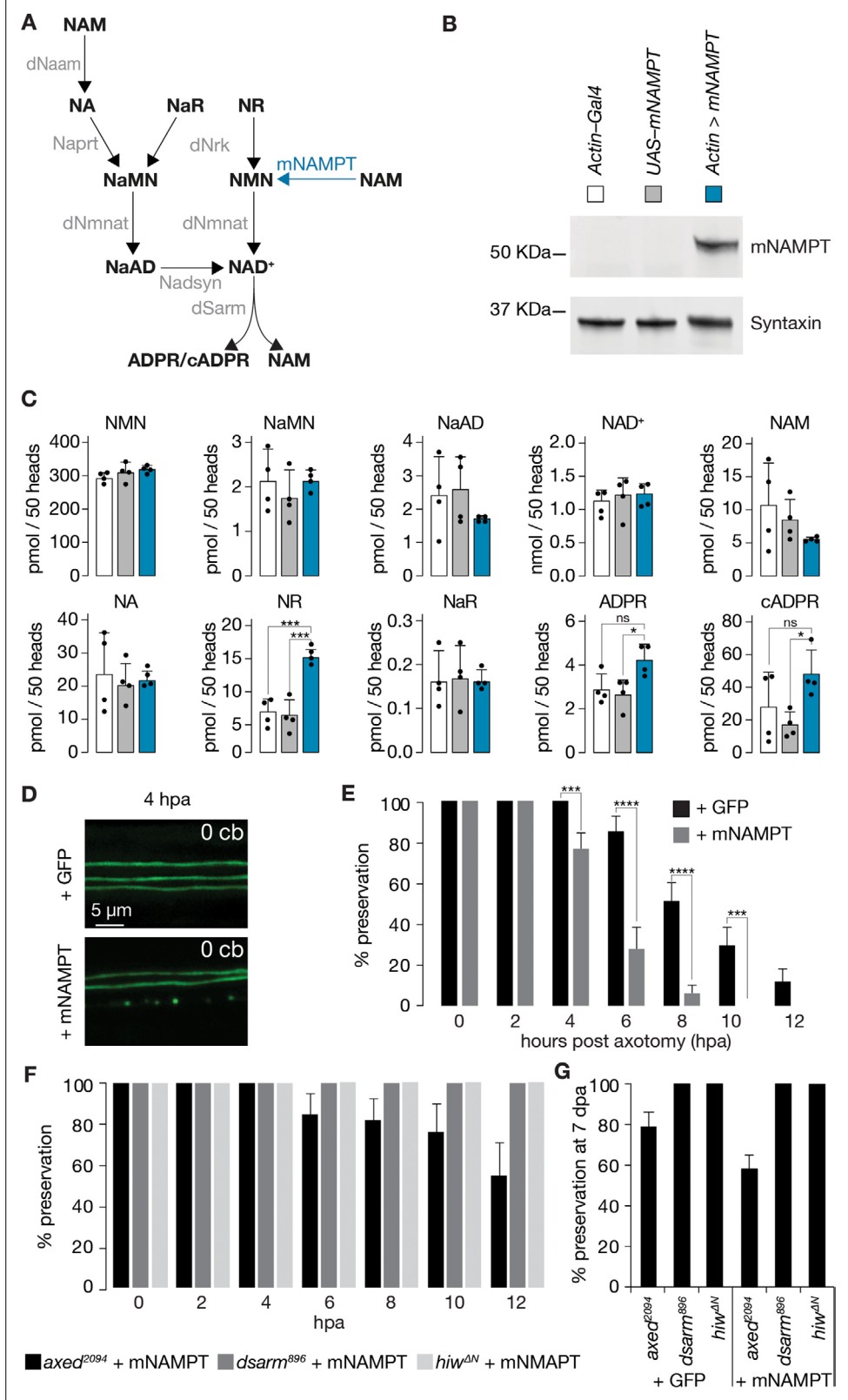

**Figure 4.** Faster injury-induced axon degeneration through mammalian NAMPT expression. (**A**) *Drosophila* NAD⁺ metabolic pathway. Black, metabolites; grey, enzymes; blue, mouse NAMPT (mNAMPT). (**B**) Detection of mNAMPT expression in heads by Western blot. (**C**) Subtle changes in NAD⁺ metabolic flux by mNAMPT expression in fly heads. Genotypes indicated in B. Extracted metabolites from 50 heads, mean ± standard deviation (n=4).

*Figure 4 continued on next page*

*Figure 4 continued*

Dots, individual measurements. One-way ANOVA with Tukey's multiple comparisons test. (**D**) The expression of mNAMPT results in faster axon degeneration after injury. Examples of injured axons at 4 hr post axotomy (hpa). (**E**) % preservation of injured axons within 12 hr post axotomy (hpa), average ± standard error of the mean (n=20 wings). Multiple unpaired t-tests. (**F**) Faster axon degeneration by mNAMPT expression requires axon death genes. % preservation of injured axons within 12 hpa, average ± standard error of the mean (n=10 wings) (**G**) % preservation of injured axons at 7 dpa, average ± standard error of the mean (n=15 wings) ****=p < 0.0001, ***=p < 0.001, *=p < 0.01, ns = p > 0.05.

The online version of this article includes the following source data and figure supplement(s) for figure 4:

**Source data 1.** Raw data of Western blot, metabolic measurements, and quantification of preserved severed axons.

**Figure supplement 1.** mNAMPT expression does not alter mRNA abundance of axon death or NAD$^+$ synthesis genes in *Drosophila* heads.

**Figure supplement 1—source data 1.** Raw data of qRT-PCR.

**Figure supplement 2.** Preservation provided by *axed* is not altered by the expression of mNAMPT.

**Figure supplement 2—source data 1.** Raw data of quantified preserved severed axons.

Next, we generated transgenic flies expressing tagged and untagged wild-type and mutant dSarm(E) variants–by targeted insertion of the *UAS–dsarm(E)* plasmids (*attP40*)–and confirmed pancellular expression of the FLAG-tagged variants by immunoblotting (***Figure 5D***). We used all variants for *dsarm*[896] axon death defective rescue experiments in our wing injury assay. We found that the expression of wild-type dSarm(E) (both tagged and untagged) almost entirely rescued *dsarm*[896] mutants, whereas dSarm(E[K450R]) proteins completely failed to rescue the phenotype at 7 dpe (***Figure 5E***). We demonstrate that a non-inducible NADase variant, dSarm(E[K450R]), in the absence of wild-type dSarm, fails to execute injury-induced axon degeneration in vivo.

## The preservation of severed axons provided by NMN-D is partially reverted by RNAi-mediated knockdown of Nadsyn

We have now established that NMN activates dSarm to trigger the degeneration of severed axons in *Drosophila*. While NMN induces a conformational change in a pocket of the ARM domain, NAD$^+$ prevents this activation by competing for the same pocket (***Bratkowski et al., 2020***; ***Figley et al., 2021***; ***Jiang et al., 2020***; ***Zhao et al., 2019***). We therefore wanted to test whether the preservation provided by lower NMN is reverted by a simultaneous reduction of NAD$^+$ synthesis. We generated NMN-D-expressing neurons containing RNAi-mediated knockdown of Nadsyn (*nadsyn*[RNAi]). At 7 dpa, the 100% preservation provided by NMN-D was partially reduced to 60% by *nadsyn*[RNAi] in vivo (***Figure 5—figure supplement 2***). This observation supports the degenerative NMN and the protective NAD$^+$ function by activating and inhibiting dSarm in injury-induced axon degeneration in *Drosophila*.

## Low NMN delays neurodegeneration triggered by loss of *dnmnat*

Lowering levels of NMN confers very robust protection against axon degeneration in *Drosophila*, similar to that achieved by targeting other mediators of axon degeneration, such as *hiw, dsarm, axed*, and the over-expression of *dnmnat* (*dnmnat*[OE]) (***Fang et al., 2012***; ***Neukomm et al., 2017***; ***Neukomm et al., 2014***; ***Osterloh et al., 2012***; ***Paglione et al., 2020***; ***Xiong et al., 2012***). We therefore assessed the genetic interaction among these regulators in vivo.

The current model, supported by our data, predicts that NMN accumulation occurs upstream of dSarm activation. Consistent with this, the induced expression of constitutively active dSarm lacking its inhibitory ARM domain (dSarm[ΔARM]) is sufficient to pathologically deplete NAD$^+$, triggering axon- and neurodegeneration in the absence of injury (***Essuman et al., 2017***; ***Neukomm et al., 2017***). We asked whether lowering levels of NMN can delay or prevent neurodegeneration induced by dSarm[ΔARM]-mediated NAD$^+$ depletion. *dsarm*[ΔARM] clones with forced NAD$^+$ depletion rapidly degenerated within 5 days after adults were born (days post eclosion, dpe) (***Figure 6A***). As expected, lowering NMN levels by NMN-D in *dsarm*[ΔARM] clones did not alter the kinetics of

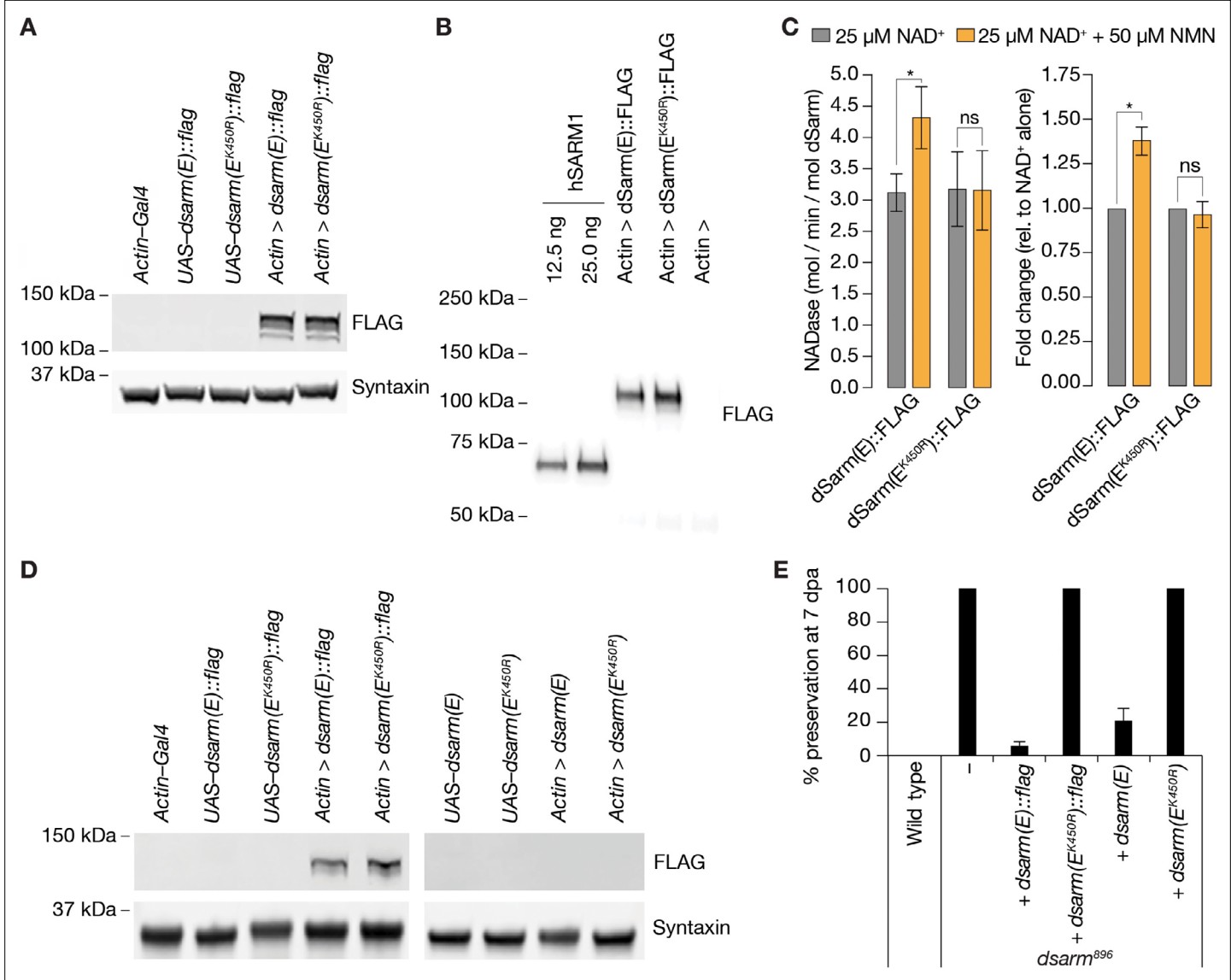

**Figure 5.** NMN inducibility of dSarm NADase is required for axon degeneration in vivo. (**A**) Expression and detection of wild-type and K450R dSarm(E) variants in S2 cells. (**B**) Immunoblot of immunopurified dSarm(E)::FLAG and dSarm(E$^{K450R}$)::FLAG. Known amounts of immunopurified human SARM1 (hSARM1) were used to quantify the levels of immunopurified dSarm(E); 12.5 ng=154.5 fmol hSARM1; 25 ng=309 fmol hSARM1 (**C**) NADase activity of dSarm. Left: NADase activity (mol NAD consumed / min / mol dSarm) of immunopurified dSarm(E)::FLAG and dSarm(E$^{K450R}$)::FLAG in the presence of 25 μM NAD$^+$, or 25 μM NAD$^+$+50 μM NMN. Right: degree of MNM induction (fold-change relative to NAD$^+$ alone). Mean ± standard error of mean (n=7). Control immunoprecipitations (using extracts from *Actin–Gal4* transfected S2 cells) revealed no non-specific NAD$^+$-consuming activity on equivalent amounts of bead / antibody complexes compared to that used in the dSarm(E) activity assays (n=7). Multiple paired t-test with false discovery rate (FDR) correction. (**D**) Equal expression levels of dSarm(E) variants in *Drosophila* heads. (**E**) Rescue experiments of dSarm(E) variants in *dsarm$^{896}$* mutant clones. dSarm(E) rescues, while dSarm(E$^{K450R}$) fails to rescue the *dsarm$^{896}$* axon death defective phenotype. % preservation of severed axons at 7 dpa, average ± standard error of mean (n=15 wings). ns = p > 0.05, *=p < 0.05.

The online version of this article includes the following source data and figure supplement(s) for figure 5:

**Source data 1.** Raw data of Western blots, NADase activity, and quantification of preserved severed axons.

**Figure supplement 1.** Partial rescue of *dsarm$^{896}$* axon death defective phenotype by dSarm(D) isoform.

**Figure supplement 1—source data 1.** Raw data of Western blots and quantification of preserved severed axons.

**Figure supplement 2.** Preservation of severed axons provided by NMN-D requires Nadsyn.

**Figure supplement 2—source data 1.** Raw data of quantification of preserved severed axons.

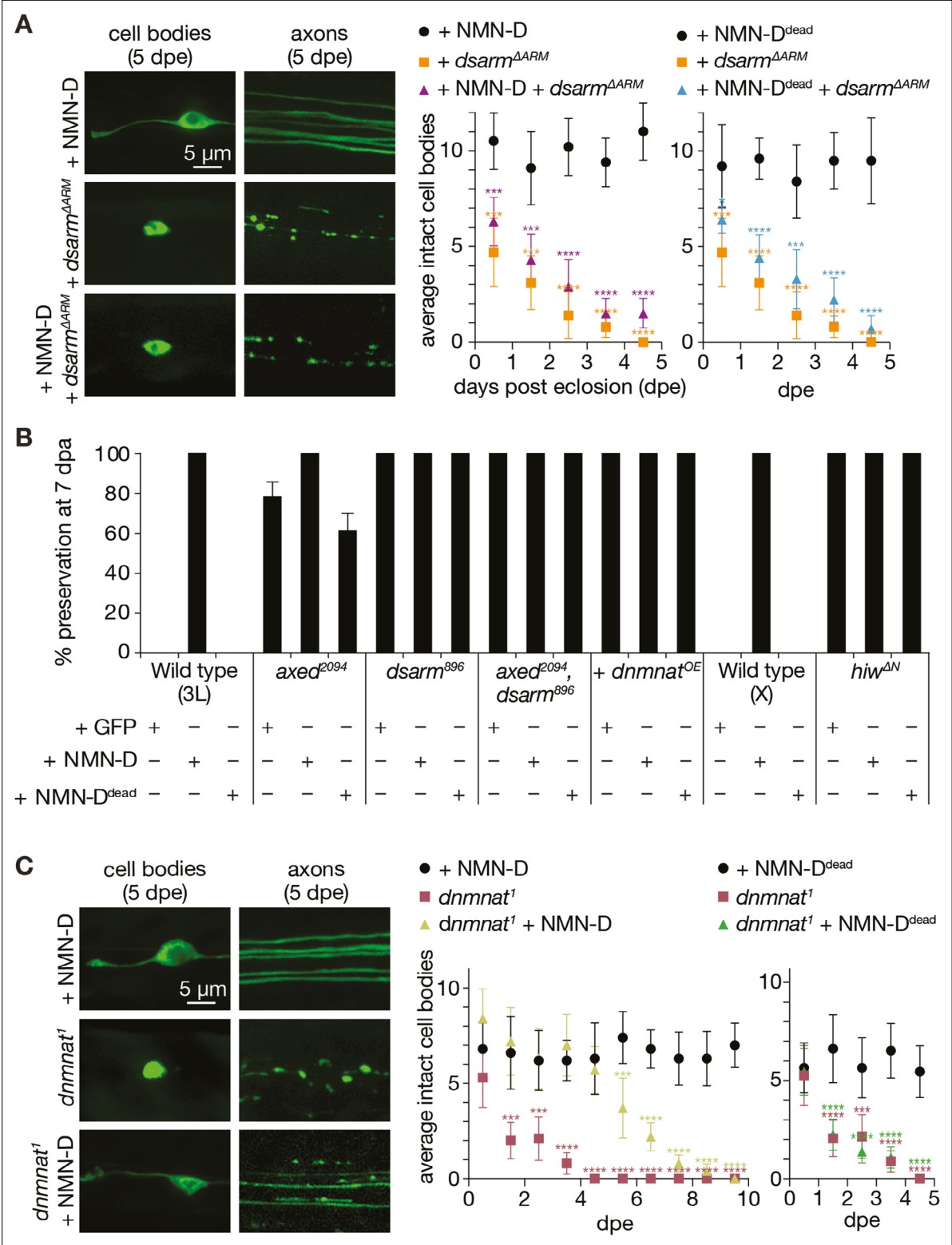

**Figure 6.** Low neuronal NMN delays neurodegeneration triggered by the loss of *dnmnat*. (**A**) Low NMN fails to prevent neurodegeneration triggered by dSarm<sup>ΔARM</sup>-mediated NAD⁺ depletion. Top: examples of cell bodies and axons at 5 days post eclosion (dpe). Bottom: quantification of intact cell bodies, average ±95% confidence interval (CI), (n=10 wings). (**B**) Epistasis analysis of low NMN with axon death signaling genes. Low NMN does not alter *axed*, *dsarm*, *hiw*, or *dnmnat* overexpression (+*dnmnat*^OE^) phenotypes in the wing injury assay. % preservation of injured axons at 7 dpa, average ± standard error of the mean (n=15 wings). (**C**) Low NMN delays neurodegeneration triggered by the loss of *dnmnat*. Top: examples of cell bodies and axons at 5

*Figure 6 continued on next page*

*Figure 6 continued*

dpe. Bottom: quantification of intact cell bodies, average ±95% CI, (n=10 wings). Multiple unpaired t-tests. All tests are compared to the control group (e.g., black dots). ****=p < 0.0001, ***=p < 0.001, *=p < 0.05.

The online version of this article includes the following source data and figure supplement(s) for figure 6:

**Source data 1.** Raw data of time course and quantification of preserved severed axons.

**Figure supplement 1.** Newly generated *sgRNAs* targeting axon death genes.

**Figure supplement 1—source data 1.** Raw data of quantification of preserved severed axons, antennal grooming, and time course.

neurodegeneration (*Figure 6A*). These observations further support that NMN accumulation occurs upstream of dSarm activation, and that once neuronal NAD+ is low, neurodegeneration cannot be halted by low NMN.

We also asked whether lowering NMN interferes with loss-of-function mutations of *axed*, *dsarm*, double mutants, *hiw*, and *dnmnat*^OE^. As expected, the attenuated axon death phenotype of low NMN did not change in these mutant backgrounds at 7 dpe (*Figure 6B*).

Loss-of-function mutations in *dnmnat* also activate axon death signaling, leading to neurodegeneration in the absence of injury (*Neukomm et al., 2017*). dNmnat is the sole enzyme with NAD+ biosynthetic activity in *Drosophila*. *dnmnat1* mutant clones lack NMN consumption and NAD+ synthesis (*Zhai et al., 2006*), and they degenerate with similar kinetics as *dsarm^ΔARM^* clones (*Neukomm et al., 2017*). We asked whether low NMN levels can delay or prevent *dnmnat1*-induced neurodegeneration. Surprisingly, although not expected to restore NAD+ synthesis, lowering NMN levels significantly delayed neurodegeneration (*Figure 6C*). Between 1–5 dpe, while clones with *dnmnat1* fully degenerated, NMN-D expressing *dnmnat1* clones remained morphologically intact, similar to controls. After 5 days, these clones gradually started to deteriorate (*Figure 6C*). This protective delay of neurodegeneration depends on lowering NMN levels, as the expression of NMN-D^dead^ completely failed to protect *dnmnat1* neurons (*Figure 6C*).

To confirm that NMN-D delays *dnmnat1*-mediated neurodegeneration, we wanted to generate tissue-specific CRISPR/Cas9 *dnmnat* knockouts. CRISPR/Cas9 tools significantly facilitate genetics in *Drosophila*. Instead of using mutants in a specific genomic locus, mutations are generated by co-expressing Cas9 and *sgRNAs*.

First, we generated transgenic flies harboring *tRNA*-flanked *sgRNAs* to target four distinct loci in *dnmnat* under the control of UAS (*Port and Bullock, 2016*). We made similar transgenic flies to target all other axon death genes (*Figure 6—figure supplement 1A*). We then tested our novel tools for their ability to attenuate axon death by assessing preserved axonal morphology (*Figure 6—figure supplement 1B*) and synaptic connectivity after injury (*Figure 6—figure supplement 1C*). We found that the preservation depended on the combination of the tissue (e.g. Gal4 driver) and the Cas9 source (e.g. *UAS–cas9* vs. *actin–cas9*). Our observations highlight that the combination of Cas9 and *sgRNAs* must be carefully determined in each tissue targeted by CRISPR/Cas9.

We then asked whether *dnmnat^sgRNAs^* can trigger neurodegeneration by analyzing neuronal survival (*Figure 6—figure supplement 1D*) and synaptic connectivity over time (*Figure 6—figure supplement 1E*). Notably, we observed synthetic lethality in *UAS–dnmnat^sgRNAs^ actin–Cas9* flies; we thus used *UAS–cas9*. Neurons with CRISPR/Cas9-targeted *dnmnat* degenerated as fast as *dnmnat1* mutants (*Figure 6—figure supplement 1D*). In line with these findings, we observed reduced synaptic connectivity in 7- and 14-day-old flies (*Figure 6—figure supplement 1E*). We also found the expression of NMN-D in *dnmnat^sgRNAs^* clones resulted in similar neuroprotection as observed with *dnmnat1* mutants. They remained morphologically intact during the first 5 days and then gradually degenerated (*Figure 6—figure supplement 1F*). Therefore, NMN-D can also delay neurodegeneration in CRISPR/Cas9-targeted *dnmnat* clones by preventing NMN accumulation.

Taken together, our in vivo results suggest that in the absence of *dnmnat,* NMN-D prevents NMN accumulation and therefore delays neurodegeneration. However, neurons subsequently degenerate because NAD+ synthesis halts, and NAD+ gradually decays below the threshold of survival. Similarly, NMN-D fails to delay neurodegeneration when dSarm^ΔARM^ forcefully depletes NAD+. These results support the role of NMN as an activator rather than an executioner in axon death signaling.

## Discussion

Here, we investigate how lowering of the NAD$^+$ precursor metabolite NMN influences axon survival in *Drosophila,* using robust expression of prokaryotic NMN-D, as demonstrated with newly generated anti-PncC/NMNd antibodies. When expressed, NMN-D consumes NMN to synthesize NaMN in *Drosophila* heads. While the preservation by NMN-D could in principle reflect an inhibitory effect of NaMN (*Sasaki et al., 2021*), the additional acceleration of degeneration by mNAMPT strongly argues that NMN is a key mediator of dSarm-driven axon degeneration. In the context of injury-induced axon degeneration, neuronal expression of NMN-D to keep NMN low is sufficient to block axon death signaling: severed axons with NMN-D remain morphologically preserved for the lifespan of flies, and circuit-integrated for weeks after injury. Notably, the NMN-D-mediated change of the NAD$^+$ metabolic flux does not alter axon death or NAD$^+$ synthesis gene expression, highlighting the usefulness of the NMN-D tool in *Drosophila*.

Neurodegeneration induced by dNmnat depletion is also delayed by low NMN levels. Our data indicate that NMN is a key mediator of axon degeneration in *Drosophila*, acting as an activator of dSarm in the axon death pathway in vivo. This is consistent with observations in mammals (*Bratkowski et al., 2020*; *Di Stefano et al., 2015*; *Zhao et al., 2019*) and with previously reported direct binding of NMN to the dSarm ARM domain (*Figley et al., 2021*).

The discovery and characterization of the axon death signaling pathway revealed four major players mediating axonal degeneration in *Drosophila*. Loss-of-function mutations in *hiw, dsarm,* and *axed*, as well as *dnmnat* over-expression robustly inhibit injury-induced axon degeneration (*Fang et al., 2012*; *Neukomm et al., 2017*; *Neukomm et al., 2014*; *Osterloh et al., 2012*; *Xiong et al., 2012*). We now show that lowering NMN levels has an equally potent protective effect, adding NMN as an additional mediator to the signaling pathway. Synaptic connectivity of severed axons is also preserved for weeks, comparable to *hiw, dsarm,* and *axed* mutants (*Neukomm et al., 2017*), and *dnmnat* over-expression (*Paglione et al., 2020*).

Our demonstration of NMN as a mediator of axon degeneration in *Drosophila* addresses an important question in the field. While recent discoveries confirm the original finding of a pro-degenerative action of NMN (*Di Stefano et al., 2015*), the role of NMN in axon degeneration has also been questioned, especially in *Drosophila*. Given the absence of NAMPT in flies, it is tempting to speculate that NMN–as a minor intermediate of the NAD$^+$ metabolic pathway–is not a primary mediator in injury-induced axon degeneration (*Gerdts et al., 2016*). However, we provide compelling evidence that NMN is not only present in flies as previously reported (*Lehmann et al., 2017*) but that its accumulation causes axon degeneration. In line with other studies, we show further proof that NMN acts as an activator of dSarm in vitro by using a new *dsarm* isoform, *dsarm(E),* which is fully functional in axon death signaling. Its non-inducible variant, dSarm(E$^{K450R}$) fails to rescue the attenuated axon degeneration phenotype in neurons lacking *dsarm*. Together with the reported dominant negative effect of SARM1($^{K193R}$) in mice (*Geisler et al., 2019*), our observations further support that NMN activation of dSarm also occurs in vivo.

The previously published NMNd revealed partially protected axons 7 days after injury in the wing (*Hsu et al., 2021*), while our NMN-D extends preservation to 50 days. This difference is likely due to the N-terminal GFP tag in GFP::NMN-D, which can increase protein stability (*Rücker et al., 2001*). This is supported by our newly generated anti-PncC antibodies and suggests that NMN-D expression levels dictate the reduction of NMN, and therefore the preservation of severed axons.

We also demonstrate that increasing the synthesis of NMN provokes a faster degeneration of severed axons in vivo, which requires all axon death mediators. Mammals synthesize NMN with two distinct enzymatic reactions: NR consumption by NRK1/2 and NAM consumption by NAMPT, both ensuring NMN supply. In *Drosophila*, NAMPT activity is absent, and NMN appears to be synthesized by Nrk alone, yet dietary NMN supplementation might also contribute to NMN levels (*Yoshino et al., 2018*). We used mNAMPT as an extra source of NMN synthesis. However, in contrast to the NMN-D-induced change in the NAD$^+$ metabolic flux, mNAMPT had only a minor impact under physiological conditions. It is challenging to measure the specific axonal rise in NMN after injury in vivo. However, NMNAT2 is rapidly disappearing in severed axons (*Gilley and Coleman, 2010*), and so is dNmnat in *Drosophila* axons and synapses (*Xiong et al., 2012*), through PHR1 and Hiw, respectively, while NAMPT persists longer (*Di Stefano et al., 2015*). It is therefore likely that persisting mNAMPT in

severed *Drosophila* axons continues NMN synthesis, leading to faster NMN accumulation, dSarm activation, and faster axon degeneration.

While NMN activates dSarm by inducing a conformational change in a pocket of the inhibitory ARM domain, NAD+ competes for the same pocket, acting as an inhibitor of dSarm activation (*Jiang et al., 2020*). Our simultaneous manipulation of NMN and NAD+ levels (by NMN-D expression and *nadsyn^RNAi*, respectively) further supports that this competition is crucial in *Drosophila* to regulate dSarm activity and, consequentially, axon degeneration after axotomy in vivo.

Finally, we expanded our investigations beyond injury, by looking at NMN in a model of neuro-degeneration. dNmnat is essential for NAD+ synthesis. While neuronal clones with mutant *dnmnat¹* start to degenerate after they are born, intriguingly, co-expression of NMN-D resulted in a preserved neuronal morphology for at least 5 days, before degeneration started with similar kinetics. Our results suggest that a rise in NMN, rather than the lack of NAD+ biosynthesis, is a trigger for neurons to degenerate also in this model, at least within the first 5 days. Once NAD+ levels drop beyond neuronal survival, neurons eventually degenerate. This is supported by observation with forced NAD+ depletion by dSarm^ΔARM (*Neukomm et al., 2017*) and inhibition of NAD+ biosynthesis in murine neurons with FK866 (*Di Stefano et al., 2015*). Still, it is surprising to observe that neurons lacking NAD+ synthesis can survive for days. It suggests either that the NAD+ turnover is slower than expected (*Liu et al., 2018*) or mechanisms are in place to compensate for NAD+ loss, at least in the short term.

In conclusion, NMN is a potent mediator of axon- and neurodegeneration in *Drosophila*. Our newly developed NMN-D tool will be useful in many degenerative aspects beyond injury, such as in axon morphogenesis and maintenance (*Izadifar et al., 2021*) and in dendrite pruning (*Ji et al., 2022*). Our metabolic analyses further demonstrate that *Drosophila* serves as an excellent model system to study NAD+ metabolism in vivo.

## Materials and methods

### Fly genetics

Flies (*Drosophila melanogaster*) were kept on Nutri-Fly Bloomington Formulation (see resources table) with dry yeast at 20 °C unless stated otherwise. The following genders were scored as progeny from MARCM crosses: females (X chromosome); and males & females (autosomes, chromosomes 2 L, 2 R, 3 L, and 3 R). We did not observe any gender-specific differences in clone numbers or axon death phenotype. Gender and genotypes are listed in *Source data 1*.

### NAD-related enzymes assay

#### Sample extraction

Fly heads (previously collected and frozen, 50 weighed heads per sample) were ground in liquid $N_2$ and sonicated after resuspension in 200–250 µl of 50 mM Tris-HCl pH 7.5, 0.3 M NaCl, 1 mM PMSF, and 2 µg/ml each of aprotinin, leupeptin, chimostatin, pepstatin and antipain. The suspension was centrifuged at 40,000 g for 20 min at 4 °C. The supernatant was passed through a G-25 column (GE Healthcare) equilibrated with 50 mM Tris-HCl pH 7.5, 0.3 M NaCl to remove low molecular weight compounds that interfere with the enzymatic assays. Protein contents were measured with the Bio-Rad Protein Assay.

#### Nampt, dNrk, Naprt, and Qaprt activities

Enzymes were assayed according to *Zamporlini et al., 2014* with minor modifications. Briefly, their formed reaction products, either NMN or NaMN, were converted to NAD using ancillary enzymes PncC (bacterial NMN Deamidase), NadD (bacterial NaMN adenylyltransferase), and NadE (bacterial NAD synthase), followed by quantification of NAD with a fluorometric cycling assay (*Zamporlini et al., 2014*).

First, mononucleotide products were converted to NaAD in dedicated assay mixtures as described below.

### Nampt
The assay mixture consisted of ethanol buffer (30 mM HEPES/KOH pH 8.0, 1% v/v ethanol, 8.4 mg/ml semicarbazide), 40 mM HEPES/KOH pH 7.5, 10 mM KF, 10 mM MgCl$_2$, 2.5 mM ATP, 0.3 mM NAM, 2 mM PRPP, 6 U/ml ADH, 0.067 mg/ml BSA, 1 U/ml NadD, 0.03 U/ml PncC, in a final volume of 100 µl.

### dNrk
The assay mixture was similar to the one of NAMPT, lacking PRPP, with 2 mM NR instead of NAM, and with 5 µM FK866.

### Naprt
The assay mixture included ethanol buffer, 40 mM HEPES/KOH pH 7.5, 10 mM KF, 20 mM MgCl$_2$, 2.5 mM ATP, 2 mM PRPP, 0.5 mM NA, 6 U/ml ADH, 0.067 mg/ml BSA and 1 U/m NadD.

### Qaprt
The assay mixture included ethanol buffer, 30 mM potassium phosphate buffer pH 7.0, 10 mM KF, 5 mM MgCl$_2$, 2.5 mM ATP, 2 mM PRPP, 0.3 mM QA, 6 U/ml ADH, 0.067 mg/ml BSA and 1 U/ml NadD.

Second, aliquots of the assay mixtures were withdrawn at different incubation times at 37 °C, treated with perchloric acid to stop the reactions, and incubated in a NadE mixture to transform NaAD into NAD. Third, NAD was quantified with fluorometric cycling (*Zamporlini et al., 2014*).

### dNaam activity
NA was converted to NaAD by the consecutive actions of the ancillary enzymes PncB (bacterial NaPRT) and NadD. The reaction mixture consisted of ethanol buffer, 40 mM HEPES/KOH pH 7.5, 10 mM KF, 10 mM MgCl$_2$, 2.5 mM ATP, 2 mM PRPP, 0.3 mM NAM, 6 U/ml ADH, 0.067 mg/ml BSA, 0.5 U/ml PncB and 1 U/ml NadD. A control mixture was prepared in the absence of NAM. The generated NaAD was converted to NAD which was quantified as described above.

### dNmnat and Nadsyn activities
Enzymatic activities were determined by directly measuring the newly synthesized NAD as follows:

### dNmnat
The assay mixture consisted of 40 mM HEPES/KOH pH 7.5, 10 mM KF, 1 mM DTT, 25 mM MgCl$_2$, 1 mM ATP, 1 mM NMN. NMN was omitted in a control mixture.

### Nadsyn
The assay mixture included 50 mM HEPES/KOH pH 7.5, 10 mM KF, 50 mM KCl, 5 mM MgCl$_2$, 4 mM ATP, 20 mM glutamine, 1 mM NaAD. NaAD was omitted in a control mixture. The dNmnat and NaDS assay mixtures were incubated at 37 °C, aliquots were withdrawn and immediately subjected to acidic treatment to stop the reaction at various times. The newly synthesized NAD was quantified as described above.

One Unit (U) above refers to the amount of enzyme that forms 1 µmol/min of product at the indicated temperature, under conditions of initial velocity, for example, less than 20% of substrate consumption. Other activity values are reported as pmol/hour/50 heads of product formed and are means ± standard deviation of two independent experiments. The ancillary bacterial enzymes PncC, NadD, and NadE were prepared as described (*Zamporlini et al., 2014*), whereas *Staphylococcus aureus* PncB was prepared according to *Amici et al., 2017*.

## NAD$^+$ metabolite quantification by *LC-MS/MS*
### Sample extraction
Fly heads (50 per sample) were extracted with 125 µl of ice-cold methanol containing stable isotope-labeled (e.g. internal standard or ISTD) metabolites. Sample extracts were vortexed and centrifuged (15 min, 14,000 rpm at 4 °C). The resulting supernatant was collected and evaporated to dryness in a vacuum concentrator (LabConco, Missouri, US). Sample extracts were reconstituted in 50 µl of ddH$_2$0 prior to LC-MS/MS analysis.

## LC-MS/MS

Extracted samples were analyzed by Liquid Chromatography coupled with tandem mass spectrometry (LC-MS/MS) in positive electrospray ionization (ESI) mode. An Agilent 1290 Infinite (Agilent Technologies, Santa Clara, California, US) ultra-high performance liquid chromatography (UHPLC) system was interfaced with Agilent 6495 LC-MS QqQ system equipped with an Agilent Jet Stream ESI source. This LC-MS/MS was used to quantify the intermediates implicated in NAD$^+$ de novo *synthesis* and *salvage pathways* (*van der Velpen et al., 2021*).

The separation of NAD$^+$ metabolites implicated in salvage and Preiss-Handler pathway was carried out using the Scherzo SMC18 (3 µm 2.0 mm x 150 mm) column (Imtakt, MZ-Analysentechnik, Mainz, Germany). The two mobile phases were composed of 20 mM ammonium formate and 0.1% formic acid in ddH$_2$O (=A) and acetonitrile: ammonium formate 20 mM and 0.1% formic acid (90:10, v/v) (=B). The gradient elution started at 100% A (0–2 min), reaching 100% B (2–12 min), then 100% B was held for 3 min and decreased to 100% A in 1 min following for an isocratic step at the initial conditions (16–22 min). The flow rate was 200 µl/min, the column temperature 30 °C and the sample injection volume 2 µl. To avoid sample carry-over, the injection path was cleaned after each injection using a strong solvent (0.2% formic acid in methanol) followed by a weak solvent (0.2% formic acid in ddH$_2$O).

AJS ESI source conditions operating in positive mode were set as follows: dry gas temperature 290 °C, nebulizer 45 psi and flow 12 l/min, sheath gas temperature 350 °C and flow 12 l/min, nozzle voltage +500 V, and capillary voltage +4000 V. Dynamic Multiple Reaction Monitoring (DMRM) acquisition mode with a total cycle of 600ms was used operating at the optimal collision energy for each metabolite transition.

## Data processing

Data was processed using Mass Hunter Quantitative (Agilent). For absolute quantification, the calibration curve and the internal standard spike were used to determine the response factor. Linearity of the standard curves was evaluated using a 14-point range; in addition, peak area integration was manually curated and corrected where necessary. Concentration of metabolites were corrected for the ratio of peak area between the analyte and the ISTD, to account for matrix effects.

## S2 cell culture

*Drosophila* Schneider cells (S2) are sold and authenticated by Thermo Fisher (R69007). The Master Seed Bank has been tested for contamination of bacteria, yeast, mycoplasma, and virus and has been characterized by isozyme and karyotype analysis. S2 cells were maintained at 26 °C in *Drosophila* Schneider's medium (Thermo Fisher) supplemented with 10% Fetal Bovine Serum (Thermo Fisher) and 1xPenicillin-Streptomycin (Thermo Fisher). 8x10$^5$ cells were plated out in 10 mm plates 24 hr prior to transfection. Cells were co-transfected with either *UAS–dsarm(E)::flag, UAS–dsarm(E$^{K193R}$)::flag, UAS–GFP::NMN-Deamidase, or UAS–GFP::NMN-Deamidase$^{dead}$* constructs and *pAc–GAL4* (Addgene) to a final concentration of 10 µg DNA/well using Mirus TransIT-Insect (Mirus Bio). Forty-eight hr post-transfection, cells were harvested with the original medium in tubes on ice. Cells were centrifuged at 5000 g for 30 s, the supernatant discarded, the cells resuspended in 5 ml of cold PBS and centrifuged again at 5000 g for 30 s. After discarding the supernatant, cells were resuspended in 300 µl/plate of cold KHM lysis buffer (110 mM CH$_3$CO$_2$K, 20 mM HEPES pH 7.4, 2 mM MgCl$_2$, 0.1 mM digitonin, Complete inhibitor EDTA free (Roche)) and incubated for 10 min at 4 °C while briefly vortexing for 5 s and up-and-down pipetting five times every minute. Samples were then centrifuged at 3000 rpm for 5 min to pellet cell debris. Protein concentration of the supernatant was determined using the BCA protein quantification assay (ThermoFischer).

## PncC antibody generation

Rabbit anti-PncC antibodies were generated by Lubioscience under a proprietary protocol. The immunogen used was purified from *Escherichia coli*, strain K12, corresponding to the full protein sequence of NMN-D. The amino acid sequence is the following:

MTDSELMQLSEQVGQALKARGATVTTAESCTGGWVAKVITDIAGSSAWFERGFVTYSNEAKAQM IGVREETLAQHGAVSEPVVVEMAIGALKAARADYAVSISGIAGPDGGSEEKPVGVWFAFATARGEGITRREC FSGDRDAVRRQATAYALQTLWQQFLQNT.

## Western blot

### Sample preparation

#### Fly heads

Whole fly heads were lysed in Laemmli buffer (2 heads/10 µl) and 10 µl loaded per well.

#### S2 cells

Protein concentration was determined as descried above, and 20 ug of protein prepared in Laemmli buffer and loaded per well.

### Sample run

Four to 12% surePAGE gels (genescript) were used with MOPS running buffer (for higher molecular weight proteins) or MES running buffer (for lower molecular weight proteins). Gels were subjected to 200 V. A molecular weight marker Precision Plus Protein Kaleidoscope Prestained Protein Ladder was used (Biorad). Proteins were transferred to PVDF membranes with the eBlot L1 system using eBlot L1 Transfer Stack supports (Genscript) and the resulting membranes were washed three times with TBS-T (Tris-buffered saline containing 0.1% Tween 20 (Merk)). Membranes were blocked with 5% milk (Carl-Roth) in TBS-T at room temperature (RT) for 1 hr. Membranes were incubated at 4 °C with corresponding primary antibodies overnight (O/N). Membranes were washed three times with TBS-T for 10 min and incubated with secondary antibodies in 5% milk in TBS-T at RT during 1 hr. Membranes were washed three times with TBS-T for 10 min.

### Antibody concentrations

#### Primary antibodies

1:5000 rabbit anti-GFP (Abcam, ab6556), 1:15,000 mouse anti-Tubulin (Sigma), 1:5000 rabbit anti-Tubulin (Abcam, T9026), 1:2000 rabbit anti-PncC (LubioScience, established in this study), 1:1000 mouse anti-FLAG (Sigma, F3165), 1:500 mouse anti-Syntaxin (DSHB, 8c3), 1:2000 anti-NAMPT (Merk, MABS465).

#### Secondary antibodies

1:10,000 goat anti-rabbit IgG (H+L) Dylight 800 (ThermoFisher, A32735), 1:10,000 goat anti-mouse IgG (H+L) Dylight 800 (ThermoFisher, A32730), 1:10,000 goat anti-rabbit IgG (H+L) Dylight 680 (ThermoFisher, A32734), 1:10,000 goat anti-mouse IgG (H+L) Dylight 680 (ThermoFisher, A32729), 1:10,000 goat anti-rat IgG (H+L) Dylight 800 (ThermoFisher, sa510024).

### Signal acquisition

Fluorescent signals were acquired using Odissey DLx (LI-COR). Images were quantified by densitometric analysis using ImageJ (NIH).

## Injury (axotomy) assays

### Wing injury

Flies were kept at 20 °C for 5–7 days prior axotomy, unless stated otherwise. Axotomy was performed using a modification of a previously described protocol (*Paglione et al., 2020*). One wing per anesthetized fly was cut approximately in the middle. The distal, cut-off part of the wing was mounted in Halocarbon Oil 27 on a microscopy slide, covered with a coverslip, and immediately used to count the amount of cut-off cell bodies (as readout for the number of injured axons) under an epifluorescence microscope. Flies were returned to individual vials. At 0, 2, 4, 6, 8, 10, and 12 hr post axotomy (hpa), or 7 days post axotomy (dpa), wings were mounted onto a slide, and imaged with a spinning disk microscope to assess for intact or degenerated axons, as well as the remaining uninjured control axons.

### Antennal ablation

Adults were aged at 20 °C for 5–7 days before performing antennal ablation (*Paglione et al., 2020*). Unilateral antennal ablation (e.g., removal of one antenna) was performed using high precision and ultra-fine tweezers, and flies returned to vial for the appropriate time. The ablation of 3rd antennal

segments did not damage the rest of the head or lead to fly mortality. At corresponding time points, adult brain dissections were performed as described (*Paglione et al., 2020*): decapitated heads were fixed in 4% formaldehyde in PTX (0.5% Triton X-100 in PBS) for 20 min, and washed 3x10 min with PTX. Brain dissections were performed in PTX, and dissected brains were fixed in 4% formaldehyde in PTX for 10 min, followed by 1 hr of blocking in 10% normal goat serum (Jackson Immuno) in PTX and an O/N incubation with the following primary antibodies at 4 °C in blocking solution: 1:500 chicken anti-GFP (Rockland), and 1:150 mouse anti-nc82 (DSHB, nc82). Brains were then washed 3x10 min with PTX at RT, and incubated with secondary antibodies in PTX at RT for 2 hr: 1:200 Dylight 488 goat anti-chicken (abcam, ab96947), and 1:200 AlexaFluor 546 goat anti-mouse (ThermoFisher, a-11030). Brains were washed 3x10 min with PTX at RT, and mounted in Vectashield for microscopy.

## Time course of degenerating neurons

Wings of aged flies (0–10 days post eclosion (dpe)) were observed and imaged with a spinning disk microscope to assess for intact or degenerated neurons and axons.

## Transgenesis

The plasmids listed below were generated and used for PhiC31 integrase-mediated targeted transgenesis (Bestgene) (5xUAS, w+ marker). *attP40* target site: *UAS–GFP::NMN-Deamidase, UAS–GFP::NMN-Deamidase^{dead}, UAS–dsarm(E), UAS–dsarm(E)::flag, UAS–dsarm(E^{K450R}), UAS–dsarm(E^{K450R})::flag, UAS–mNAMPT. VK37* target site: *UAS–4 x(tRNA::axed^{sgRNAs}), UAS–4 x(tRNA::hiw^{sgRNAs}), UAS–4 x(tRNA::dsarm^{sgRNAs}), UAS–4 x(tRNA::dnmnat^{sgRNAs})*. All plasmids are available as *.gb files on Addgene.

## Optogenetics

Crosses were performed on standard cornmeal agar containing 200 µM all-*trans* retinoic acid in aluminum-wrapped vials to keep the progeny in the dark (*Paglione et al., 2020*). Adult progenies were aged at 20 °C for 7-14 days before starting the experiment. CsChrimson experiments were performed in the dark, and flies were visualized for recording using an 850 nm infrared light source at 2 mW/cm² intensity (Mightex, Toronto, CA). For CsChrimson activation, 656 nm red light at 6 mW/cm² intensity (Mightex) was used. Red light stimulus parameters were delivered using a NIDAQ board controlled through Bonsai (https://open-ephys.org/). Exclusion criteria: to avoid spontaneous grooming behavior, during the recording, flies that groomed within the first 30 s were excluded from the analysis. Red-light stimulation (10 Hz for 10 s) was followed by a 30 s interstimulus recovery (3 repetitions in total). Flies were recorded, and videos were manually analyzed using VLC player (http://www.videolan.org/). Grooming activity (ethogram) was plotted as bins (1 bin, grooming event(s) per second). Ethograms were visualized using R (https://cran.r-project.org/). The ablation of 2nd antennal segments did neither damage the head nor lead to fly mortality. Flies that died during the analysis window (7–15 dpa) were excluded.

## In-cell NAD-glo of dSarm proteins for NADase assay

### Immunoprecipitation

S2 cells cell lysates (see above) were protein-quantified with the BCA protein assay and diluted to 500 ng/µl in ice-cold KHM buffer. Lysates were mixed with 20 µg/ml mouse anti-FLAG M2 monoclonal antibody (Sigma-Aldrich, F3165) and 50 µl/ml of pre-washed Pierce magnetic protein A/G beads (Thermo Fisher Scientific, 88802) and incubated overnight at 4 °C with rotation. After incubation, beads were washed 3 x with KHM buffer and 1 x with PBS and resuspended in 1 mg/ml BSA in PBS (with protease inhibitors, Merk, 11873580001).

### NADase assays

A series of test assays were first performed to define appropriate test conditions. Optimized reaction conditions were as follows: 25 µl reactions (overall 1 x PBS) contained 40 fmol/µl dSarm(E) protein together with 25 µM NAD⁺±50 µM NMN. Reactions were kept on ice while being set up. Reactions were performed with the recombinant dSarm(E) still attached to beads and bead suspensions were thoroughly mixed prior to addition to the reactions. Constitutive (basal) NAD⁺ consumption was measured from reactions containing NAD⁺ alone as the difference between starting levels (0 mins) and levels remaining after incubating for between 80 and 120 min at 25 °C, and NAD⁺ consumption in

the presence of 50 µM NMN was calculated after incubating for between 40 and 120 min (times were dependent on variant activity in each sample). Reactions were mixed once during the incubation to resuspend the beads. $NAD^+$ levels were measured using the NAD/NADH-Glo assay. Five µl aliquots of reaction were removed immediately after setting up (whilst still on ice), to obtain precise starting levels (0 min) in individual reactions, and again after the defined times listed above. Aliquots were then mixed with 2.5 µl of 0.4 M HCl, to stop the reaction, and neutralised by mixing with 2.5 µl 0.5 M Tris base after 10 min. Neutralised samples were subsequently diluted 1 in 50 in a buffer consisting of 50% PBS, 25% 0.4 M HCL, 25% 0.5 M Tris base to bring the $NAD^+$. $NAD^+$ concentrations down to the linear range of detection for the NAD/NADH-Glo assay. Ten µl of the diluted sample was then mixed with 10 µl of NAD/NADH-Glo detection reagent on ice in wells of a 384-well white polystyrene microplate (Corning). Once all reactions had been set up the plate was moved to a GloMax Explorer plate reader (Promega) and incubated for 40 min at 25 °C before reading for luminescence. $NAD^+$ concentrations were determined from a standard curve generated from a dilution series of $NAD^+$ and $NAD^+$ consumption rates were converted to mol of $NAD^+$ consumed per min per mol of dSarm(E) protein (mol/min/mol dSarm) (*Gilley et al., 2021*). Individual data points for each separate protein preparation are the means of two or three technical replicates. No non-specific activity was detected on bead/antibody complexes in control immunoprecipitations using extracts from *Actin–Gal4* transfected S2 cells (based on n=5).

## Quantitative PCR with reverse transcription and RNA quantification

Total RNA from forty fly heads for each genotype was isolated with TRIzol LS Reagent (Invitrogen, 10296010). The isolated RNA was treated with TURBO DNase (Invitrogen, AM2238) at 37 °C for 20 min and purified using RNA Clean & Concentrator-5 (Zymo Research, R1015). First strand cDNA was synthesized using random hexamers (Invitrogen, N8080127) and SuperScript IV first-strand synthesis system (Invitrogen, 18091050). Quantitative PCR was performed for each sample using PowerUp SYBR green master mix (Applied Biosystems, A25741) with technical triplicates for both +RT and −RT for each genotype in MicroAmp optical 96-well reaction plates (Applied Biosystems, 4306737) and QuantStudio 1 real-time PCR system (Applied Biosystems). Relative transcript abundance was calculated using the ΔΔCt method. α-tubulin, an mRNA that remains unchanged in mutant and transgenic flies based on qPCR analysis, was used for normalization. Statistical significance was calculated using the one-way ANOVA test. The post hoc Tukey test was performed for statistically different groups determined by the ANOVA test (p-value <0.05). The resulting p values from the Tukey test are reported.

Primers used (forward and reverse, respectively, 100–200 bp amplicons, all isoforms included):

*NMN-D*: TCGTGCTGATTATGCCGTGT, AAAAGCAAACCAGGTGC
*mNAMPT*: TGGGGTGAAGACCTGAGACA, TGGCAGCAACTTGTAGCCTT
*dsarm*: AGGAGAACATGGCCAAGACG, GTTGTCAATTGCCCGCCT
*axed*: CATTCCCTACCGCGCTCACA, TTTGGTGCTGGTTGGTCAGT
*hiw*: CTCACCCAGCGTCAGAAGTT, CCATTGGCTCCAATCCAGGT
*dnmnat*: TTGCTGTTCCAGGCCTATGG, CAACGTGGAGCTCACCTCAT
*dnaam*: CAATGGACGCCTGTTTCACG, TTCGTATCGAAGGCGAA
*dnrk*: GCGTGTCCCATGGAGCAATA, AGCCACGATTCGGAGAAGTA
*naprt*: TCCTATGCCATCGCATTCCC, TTGGCGGACTGTTCTCAGAC
*nadsyn*: AATATGCTGGTGGACGTGGG, GGCGATTAAAGAACGCCACC

## Replicates

For all experiments, at least 3 biological replicates were performed for each genotype and/or condition. No inclusion/exclusion criteria was applied except for optogenetics, which is stated in the subsection 'optogenetics' of Materials and methods.

## Software and statistics

Image-J and photoshop was used to process wing and ORN pictures. Software for optogenetics is included in the optogenetics section. Graphpad prism 9 was used to perform all the statistical analysis. For tests applying a false discovery rate (FDR) correction, the adjusted p value we report is the q value.

## Source data

The following R code was used to generate ethograms from excel flies.

```
library(readxl)
nmn_7 <- read_excel("nmn 7 days.xlsx")
head(nmn_7)
# custom function using image to emulate an ethograph
ethogram <-
function(zeroOneMatrix, color='skyblue',xlab='behaviour',ylab='animals'){
m <- as.matrix(zeroOneMatrix)
m[m==0] <- NA
nAnimals <- nrow(m)
nTimeSlots <- ncol(m)
image(x=1:nTimeSlots,  y=1:nAnimals,
   z=t(m[nAnimals:1,]),
   col = c(color),
   xlab = xlab,
   ylab = ylab,
   yaxt = 'n')
}
# let's plot
ethogram(nmn_7, color='lightskyblue1')

data_t=t(nmn_7)
head(data_t)
colnames(data_t)=rev(c(1:ncol(data_t)))
rownames(data_t)=c(1:nrow(data_t))
head(data_t)
data_long <- reshape2::melt(data_t)
colnames(data_long)=c("Time", "Animal", "Val")
data_long$Val <- factor(data_long$Val)
head(data_long)

data_zeros =data.frame(data_t)
data_zeros[is.na(data_zeros)] <- 0
rs = rowSums(data_zeros)
rs
data_sum = data.frame(Time = c(1:length(rs)), Count = rs)
head(data_sum)
library(ggplot2)
e2=ggplot()+
 geom_tile(data_long, mapping = aes(x=data_long$Time, y=data_long$Animal,
fill = data_long$Val), color="white", size = 0.5)+
 labs(x="behaviour", y="animals", title="OPTO")+
 theme_bw()+theme(axis.text.x=element_text(size = 9, angle = 0, vjust =
0.3),
   axis.text.y=element_text(size = 9),
   plot.title=element_text(size = 11))+
```

```
theme(panel.border=element_blank())+
scale_fill_manual(values = c("skyblue"))+
geom_line(data_sum, mapping = aes(x=data_sum$Time, y=data_sum$Count),
color="black", size = 0.7)
e2
```

## Transparent reporting guidelines

We followed the ARRIVE guidelines for reporting work involving fly research.

## Acknowledgements

We thank Dr. Jemeen Sreedharan for support in generating transgenic flies, and Dr. Julijana Ivanisevic and Dr. Hector Gallart-Ayala from the Metabolomics Unit at University of Lausanne for metabolic analyses. This work was supported by a UK Biotechnology and Biological Sciences Research Council (BBSRC) / AstraZeneca Industrial Partnership award (BB/S009582/1 a) to JG; funds from the Italian Grant RSA 2018–20 from UNIVPM to GO; a Sir Henry Wellcome Postdoctoral Fellowship from the Wellcome Trust (210904/Z/18/Z) to AL; the John and Lucille van Geest Foundation to MPC; and Swiss National Science Foundation SNSF Assistant Professor awards (PP00P3_176855 and PP00P3_211015), the International Foundation for Research in Paraplegia (P180), and SNSF Spark (190919) to LJN.

## Additional information

### Funding

| Funder | Grant reference number | Author |
| --- | --- | --- |
| Schweizerischer Nationalfonds zur Förderung der Wissenschaftlichen Forschung | 176855 | Lukas Jakob Neukomm |
| Schweizerischer Nationalfonds zur Förderung der Wissenschaftlichen Forschung | 211015 | Lukas Jakob Neukomm |
| Biotechnology and Biological Sciences Research Council | BB/S009582/1a | Jonathan Gilley |
| International Foundation for Research in Paraplegia | P180 | Lukas Jakob Neukomm |
| Schweizerischer Nationalfonds zur Förderung der Wissenschaftlichen Forschung | 190919 | Lukas Jakob Neukomm |
| Università Politecnica delle Marche | 2018-20 | Giuseppe Orsomando |
| Wellcome Trust | 210904/Z/18/Z | Andrea Loreto |
| John and Lucille Van Geest Foundation | | Michael P Coleman |
| Schweizerischer Nationalfonds zur Förderung der Wissenschaftlichen Forschung | 201535 | Pei-Hsuan Wu |

| Funder | Grant reference number | Author |
| --- | --- | --- |

The funders had no role in study design, data collection and interpretation, or the decision to submit the work for publication. For the purpose of Open Access, the authors have applied a CC BY public copyright license to any Author Accepted Manuscript version arising from this submission.

## Author contributions

Arnau Llobet Rosell, Conceptualization, Resources, Formal analysis, Validation, Investigation, Visualization, Methodology, Writing – original draft, Writing – review and editing; Maria Paglione, Giulia Perillo, Formal analysis, Investigation, Methodology; Jonathan Gilley, Formal analysis, Validation, Investigation, Methodology, Writing – review and editing; Magdalena Kocia, Massimiliano Gasparrini, Lucia Cialabrini, Nadia Raffaelli, Carlo Angeletti, Investigation, Methodology; Giuseppe Orsomando, Investigation, Methodology, Writing – review and editing; Pei-Hsuan Wu, Investigation, Visualization, Methodology, Writing – review and editing; Michael P Coleman, Project administration, Writing – review and editing; Andrea Loreto, Conceptualization, Supervision, Validation, Writing – original draft, Writing – review and editing; Lukas Jakob Neukomm, Conceptualization, Supervision, Funding acquisition, Validation, Visualization, Writing – original draft, Project administration, Writing – review and editing

## Author ORCIDs

Arnau Llobet Rosell (ORCID) http://orcid.org/0000-0001-7728-2999
Maria Paglione (ORCID) http://orcid.org/0000-0002-0921-940X
Jonathan Gilley (ORCID) http://orcid.org/0000-0002-9510-7956
Magdalena Kocia (ORCID) http://orcid.org/0000-0002-7805-1970
Giuseppe Orsomando (ORCID) http://orcid.org/0000-0001-6640-097X
Pei-Hsuan Wu (ORCID) http://orcid.org/0000-0002-6690-0744
Andrea Loreto (ORCID) http://orcid.org/0000-0001-6535-6436
Lukas Jakob Neukomm (ORCID) http://orcid.org/0000-0002-5007-3959

## Decision letter and Author response

Decision letter https://doi.org/10.7554/eLife.80245.sa1
Author response https://doi.org/10.7554/eLife.80245.sa2

# Additional files

## Supplementary files
• MDAR checklist

• Source data 1. Genotypes in each display item. Abbreviations: mCD8::GFP = UAS–mCD8::GFP, dpr1=dpr1–Gal4

## Data availability
Generated plasmids have been deposited in Addgene. All data generated or analyzed during this study are included in the manuscript and supporting files.

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
