## [Editor Report]

The paper by Llobet Rosell et al. and the tools they develop will be valuable to the neurodegeneration/axon injury field. The authors systematically analyze the signaling role of NMN, a NAD precursor metabolite in activating the axon degeneration trigger dSARM, in *Drosophila*. They show the extent to which the NMN/NAD+ ratio drives pathological axon degeneration and demonstrate convincingly, that reducing NMN levels is strongly protective in several in vivo injury paradigms.

---

## [Decision Letter]

**Decision letter after peer review:**

Thank you for submitting your article "The NAD+ precursor NMN activates dSarm to trigger axon degeneration in *Drosophila*" for consideration by *eLife*. Your article has been reviewed by 3 peer reviewers, and the evaluation has been overseen by a Reviewing Editor and Claude Desplan as the Senior Editor. The following individual involved in the review of your submission has agreed to reveal their identity: Pete Williams (Reviewer #2).

Essential revisions:

1) Clarification of arguments and outcomes of your experiments:

a. Please see Reviewer 1's comments 1. and 3., on why low NMN should provide stronger protection than the loss of dSARM, and on the distinction between the behavior of axed and hiw/dSARM, respectively.

b. Reviewer 3: Better characterize the conclusions regarding NMN since the effects are so strong. Please characterize this in more detail. For example, the NAD concentration in axons should still decline after injury because NMNAT is degraded. Why are the axons not dying? Does NMN also affect NMNAT stability? Or is the NAD present in the axons before injury so stable it can provide for the lifetime?

2) Quantification:

a. Reviewer 1: In Figure 1E, the authors demonstrate protection at 10, 30, and 50 dpa but the data is not quantified. These data should be quantified, especially given the long-lived claims mentioned above.

b. Reviewer 1: The authors should make the distinction between the behavior of axed and hiw/dSARM clear in the text and run statistical comparisons among the groups.

3) Reviewer 2: Test the opposite of their experiment to produce prolonged reduced levels of NMN, that is, whether high levels of NMN trigger axon degeneration. You introduced the murine version of NAMPT (mNAMPT) which catalyzes NAM to NMN.

Reviewer 2 queries whether this is through the hypothesized mechanism, as mNAMPT fails to increase NMN levels, and suggests that you try to demonstrate a mechanism in which high NMN triggers axon degeneration: e.g. increasing NR or NRK whilst decreasing downstream enzymes. One hypothesis is that NMN levels are not critically reliant on mNAMPT levels until there has been a neurodegenerative injury. A (seemingly) easy way to test this is to assess NMN levels in the mNAMPT transgenic and control pre- and post-injury, to interrogate whether dynamics of the pathway results in differential regulation of the genes/proteins in the pathway. Targeted protein or gene profiling (e.g. simple qPCR or WB) would help assess this. This should not be too difficult unless you argue otherwise.

4) Description of the generation/availability of the novel/new PncC antibody mentioned.

5) Include data on the Olfactory receptor neuron degeneration assays shown in Figure 1.

6) Clarify Metabolome data:

a. Reviewer 2 in his "Weaknesses" points: you use metabolic profiling to display individual metabolites during axon degenerative events and treatments, but it is unclear if any of these proteins or genes change as a consequence. The inclusion of data on this would be helpful in explaining the mNAMPT data.

b. Reviewer 3: Please combine your metabolome (not very well explained in the manuscript) and genetic analyses and test a number of predictions from the pathway diagrams as well. For example, does the NaDS pathway become essential in NMN-D overexpressing flies?

*Reviewer #1 (Recommendations for the authors):*

1. The claim that the protection shown here is stronger than that observed with the loss of other mediators of axon degeneration (e.g. dSARM) should be removed. While it is true that the authors here do a good job of demonstrating long-lived protection, there is no reason why low NMN should provide stronger protection than the loss of dSARM.

2. In Figure 1E, the authors demonstrate protection at 10, 30, and 50 dpa but the data is not quantified. These data should be quantified, especially given the long-lived claims mentioned above.

3. In Figure 4F, the protection observed in dSARM and hiw backgrounds is insensitive to increased mNAMPT expression, as expected since increased mNAMPT is predicted to accelerate degeneration in a dSARM-dependent manner. However, axed-mediated protection is sensitive to elevated mNAMPT arguing that axed acts either upstream or in parallel to NMN levels. These data are not consistent with Axed acting downstream of dSARM. The authors should make the distinction between the behavior of axed and hiw/dSARM clear in the text and run statistical comparisons among the groups.

4. It appears that the data shown in Figures 4F and 4G are inconsistent. The authors show that mNAMPT overexpression in an axed background gives 50% protection at both 12 hours and 7 dpa.

*Reviewer #2 (Recommendations for the authors):*

In general, the manuscript by Llobet Rosell and colleagues is well written and well presented. The data is mostly clear and supportive of the primary hypotheses with a few exceptions mentioned below. As *Drosophila* is a clear model organism to be used for axon degenerative research, the data and tools presented here should be beneficial to the neuroscience community. As there is now mounting evidence that high NMN (or high ratios of NMN to NAD) triggers axon degeneration, the data presented here is timely and should be generally well-received.

The following comments should be addressed to support the hypotheses presented in the manuscript:

1) In this manuscript the authors expertly use the prokaryotic enzyme NMN-D to deplete NMN, with a stabilized version allowing for prolonged NMN depletion. This strongly supports the role of low NMN in axon survival. The authors next question whether high levels of NMN trigger axon degeneration. To test this hypothesis they introduce the murine version of NAMPT (mNAMPT) which catalyzes NAM to NMN. The neurodegenerative outcomes support their hypothesis but it is unclear whether this is through the hypothesized mechanism as mNAMPT fails to increase NMN levels. To support this hypothesis the authors need to demonstrate a mechanism in which high NMN triggers axon degeneration: e.g. increasing NR or NRK whilst decreasing downstream enzymes. Is there a fly bioavailable version of NMN that could be added directly to diet? One hypothesis could be that NMN levels are not critically reliant on mNAMPT levels until there has been a neurodegenerative injury. A (seemingly) easy way to test this is to assess NMN levels in the mNAMPT transgenic and control pre- and post-injury. It could also be that changing the dynamics of the pathway results in differential regulation of the genes/proteins in the pathway. Targeted protein or gene profiling (e.g. simple qPCR or WB) would help assess this. As a minor comment, it is also unclear as to why murine NAMPT was chosen and not another orthologue (as this could explain why the enzyme doesn't seem to be active as predicted). This justification can be addressed within the text.

2) The authors introduce a novel/new PncC antibody however there are no details on this, its generation, availability, or how it was tested. This should be included in the manuscript. The antibody detects with several bands. The authors speculate that this could be a degradation product but nothing substantial is shown – can the authors support this claim more?

3) Olfactory receptor neuron degeneration assays are shown in Figure 1(D and E) but no data is presented alongside this to back it up. The data for these images should be presented as well.

*Reviewer #3 (Recommendations for the authors):*

Overall, the manuscript describes a large set of reagents for the study of NAD metabolism in injury-induced degeneration. In my opinion, the conclusions regarding NMN are highly interesting because the effects are so strong. In fact, I think the authors could characterize this in a bit more detail. For example, the NAD concentration in axons should still decline after injury because NMNAT is degraded. Why are the axons not dying? Does NMN also affect NMNAT stability? Or is the NAD present in the axons before injury so stable it can provide for the lifetime?

The authors could also combine their metabolome (which is actually not very well explained in the manuscript text) and genetic analyses and test a number of predictions from the pathway diagrams as well. For example, does the NaDS pathway become essential in NMN-D overexpressing flies?

Writing – could be more confident/straightforward. Examples

– "It remains currently unclear whether NMNd expression levels determine the extent of preservation." Does not sound like the best justification for an extensive study

– "Our observations highlight that the combination of Cas9 and sgRNAs has to be carefully determined in each tissue targeted by CRISPR/Cas9." Every fly researcher knows this by now…

– The introduction makes it sound as if NMNAT2 degradation is induced by injury. Is that the case? I thought it was constantly degraded and needs to be replenished.

– Are axed and hiw explained properly in the introduction?

---

## [Author Response]

Reviewer #1 (Recommendations for the authors):1. The claim that the protection shown here is stronger than that observed with the loss of other mediators of axon degeneration (e.g. dSARM) should be removed. While it is true that the authors here do a good job of demonstrating long-lived protection, there is no reason why low NMN should provide stronger protection than the loss of dSARM.

We agree with reviewer #1 that we don’t provide substantial evidence to support that low NMN offers stronger protection than the loss of other axon death genes.

In the abstract, we replaced the “or even stronger than” with “to” in the last sentence:

“The potent neuroprotection by reducing NMN levels is similar to the interference with other essential mediators of axon degeneration in *Drosophila*.”

In the discussion, we changed the text as indicated below:

“We now show that lowering NMN levels has an equally potent protective effect, adding NMN as an additional mediator to the signaling pathway. Synaptic connectivity of severed axons is also preserved for weeks, comparable to *hiw, dsarm* and *axed* mutants (Neukomm et al., 2017), and *dnmnat* over-expression (Paglione et al., 2020).”

2. In Figure 1E, the authors demonstrate protection at 10, 30, and 50 dpa but the data is not quantified. These data should be quantified, especially given the long-lived claims mentioned above.

We agree with reviewer #1 that a quantification would support our observation. However, it is difficult to precisely quantify individual axons in the ORN injury assay, for two main reasons:

1. Severed axons are often bundled, thus the exact number cannot be scored.

2. Due to the removal of the cell body, the axonal GFP intensity decreases over time, due to the absence of mCD8::GFP synthesis. It adds another level of difficulty.

Nevertheless, we added numbers to each example in Figure 1E and D, where we quantified the % of brains where severed preserved axons were observed, similar to Figure 2 in (MacDonald et al., 2006).

In the Results section, we changed the text as indicated below:

“We extended the ORN injury assay and found preservation at 10, 30, and 50 dpa (Figure 1E). While quantifying the precise number of axons is technically not feasible, severed preserved axons were observed in all 10, 30, and 50 dpa brains, albeit fewer at later time points (MacDonald et al., 2006). Thus, high levels of NMN-D confer robust protection of severed axons for multiple neuron types for the entire lifespan of *Drosophila*.”

In the Figure 1 legend, we changed the text accordingly:

“D Low NMN results in severed axons of olfactory receptor neurons that remain morphologically preserved at 7 dpa. Examples of control and 7 dpa (arrows, site of unilateral ablation). Lower right, % of brains with severed preserved axon fibers. **E** Low NMN results in severed axons that remain morphologically preserved for 50 days. Representative pictures of 10, 30, and 50 dpa, from a total of 10 brains imaged for each condition (arrows, site of unilateral ablation). Lower right, % of brains with severed preserved axon fibers.”

3. In Figure 4F, the protection observed in dSARM and hiw backgrounds is insensitive to increased mNAMPT expression, as expected since increased mNAMPT is predicted to accelerate degeneration in a dSARM-dependent manner. However, axed-mediated protection is sensitive to elevated mNAMPT arguing that axed acts either upstream or in parallel to NMN levels. These data are not consistent with Axed acting downstream of dSARM. The authors should make the distinction between the behavior of axed and hiw/dSARM clear in the text and run statistical comparisons among the groups.

We added data that helps clarifying the misunderstanding. We already observe with *axed^2094^* that a small fraction of axons undergoes degeneration within the first 12 hours post axotomy (hpa). We do not observe further degeneration after 12 hours. At 7 days post axotomy (dpa), the number of preserved axons remain unchanged. We added Figure 4 —figure supplement 2, where the preservation of severed *axed^2094^* axons ± mNAMPT expression is documented, within 12 hpa, and 7 dpa (see next page). The phenotype of *axed* mutants does not change with the expression of mNAMPT. At 7 dpa, the two groups are not significantly different (ordinary one-way ANOVA with Tukey’s multiple comparisons, ns = 0.0595). It suggests that *axed* is also downstream of NMN levels mediated by mNAMPT expression.

In the Results section, we changed the text accordingly:

“We next asked whether the faster degeneration of mNAMPT-expressing severed axons requires axon death genes. While mutations in *dsarm* and *hiw* completely blocked the degeneration of severed axons expressing mNAMPT, *axed* showed a partial preservation of 60% at 12 hours after injury (Figure 4F). Importantly, *axed* mutants, in the absence of mNAMPT expression, showed a similar preservation within the first 12 hours (Figure 4 —figure supplement 2). This preservation remained unchanged at 7 dpa, suggesting that mNAMPT expression does not change the preservation provided by *axed, dsarm,* and *hiw* (Figure 4 —figure supplement 2, Figure 4G). Our observations support that elevated NMN levels require axon death signaling to initiate the degeneration of severed axons.”

4. It appears that the data shown in Figures 4F and 4G are inconsistent. The authors show that mNAMPT overexpression in an axed background gives 50% protection at both 12 hours and 7 dpa.

See answer to point 3. above.

Reviewer #2 (Recommendations for the authors):In general, the manuscript by Llobet Rosell and colleagues is well written and well presented. The data is mostly clear and supportive of the primary hypotheses with a few exceptions mentioned below. As *Drosophila* is a clear model organism to be used for axon degenerative research, the data and tools presented here should be beneficial to the neuroscience community. As there is now mounting evidence that high NMN (or high ratios of NMN to NAD) triggers axon degeneration, the data presented here is timely and should be generally well-received.The following comments should be addressed to support the hypotheses presented in the manuscript:1) In this manuscript the authors expertly use the prokaryotic enzyme NMN-D to deplete NMN, with a stabilized version allowing for prolonged NMN depletion. This strongly supports the role of low NMN in axon survival. The authors next question whether high levels of NMN trigger axon degeneration. To test this hypothesis they introduce the murine version of NAMPT (mNAMPT) which catalyzes NAM to NMN. The neurodegenerative outcomes support their hypothesis but it is unclear whether this is through the hypothesized mechanism as mNAMPT fails to increase NMN levels. To support this hypothesis the authors need to demonstrate a mechanism in which high NMN triggers axon degeneration: e.g. increasing NR or NRK whilst decreasing downstream enzymes. Is there a fly bioavailable version of NMN that could be added directly to diet? One hypothesis could be that NMN levels are not critically reliant on mNAMPT levels until there has been a neurodegenerative injury. A (seemingly) easy way to test this is to assess NMN levels in the mNAMPT transgenic and control pre- and post-injury. It could also be that changing the dynamics of the pathway results in differential regulation of the genes/proteins in the pathway. Targeted protein or gene profiling (e.g. simple qPCR or WB) would help assess this. As a minor comment, it is also unclear as to why murine NAMPT was chosen and not another orthologue (as this could explain why the enzyme doesn't seem to be active as predicted). This justification can be addressed within the text.

We will answer point-by-point to the major comments raised in the paragraph above.

Alternative ways to elevate NMN:

1) We performed additional experiments where we fed flies overexpressing dNrk::FLAG or mNRK1::FLAG with NR, with the goal to bolster NMN. First, we were able to detect the expression of both FLAG-tagged enzymes. Second, we measure metabolites in whole heads, but no significant change in the NAD^+^ metabolism was observed. Third, we applied the wing injury assay, but no change in axon degeneration was observed within the first 12 hours. Unfortunately, by manipulating dNrK/mNRK1 combined with dietarily supplemented NR, we were not able to observe faster degeneration (data not shown).

2) We repeated the same approach with NAM and mNAMPT. However, no further changes were observed (data not shown).

Our above approaches led us to conclude that the sole expression of mNAMPT is sufficient, after injury, to provide an additional source of NMN synthesis. It is robust enough to generate a modest additional amount of NMN (but not NAD^+^), but not as massive that under physiological condition, a significant increase of NMN (and NAD^+^) results in the opposite phenotype as observed in mammals.

In the Results section, we included the following conclusion:

“Strikingly, in our wing injury assay, while axons with GFP showed signs of degeneration starting from 6 hours post axotomy (hpa), mNAMPT expression resulted in significantly faster axon degeneration with signs of degeneration at 4 hpa (Figure 4D, E). This accelerated degeneration is likely linked to increased NMN production, but other mechanisms cannot be excluded as there is no increase in NMN under physiological conditions.”

Measuring NMN pre- and post-injury:

Unfortunately, we cannot measure NMN in severed axons in the head of *Drosophila* after injury. They are surrounded by intact axons and glial cells.

Targeted gene profiling:

We added three qRT PCR figures (Figure 2 —figure supplement 1, Figure 2 —figure supplement 2, Figure 4 —figure supplement 1). The first indicates the detection of all transcripts (e.g., axon death and NAD^+^ synthesis genes), the second demonstrates that neither of them is drastically changed with NMN-D or NMN-D^dead^ expression. The third demonstrates that neither of them is drastically changed with mNAMPT expression.

We changed the results accordingly (Figure 2 —figure supplement 1):

“Before measuring the effect of NMN-D on the NAD^+^ metabolic flux, we measured the activities of the various NAD^+^ biosynthetic enzymes in fly heads (Figure 2 —figure supplement 1A, Figure 2A) (Amici et al., 2017; Zamporlini et al., 2014). We confirmed that NAD^+^ can be synthesized from nicotinamide (NAM), nicotinamide riboside (NR), and quinolinate. However, the *Drosophila* Qaprt homolog that catalyzes the conversion from quinolinate to nicotinic acid mononucleotide (NaMN) remains to be identified (Katsyuba et al., 2018). We also confirmed the absence of NAMPT activity (Gossmann et al., 2012) (Figure 2 —figure supplement 1A). In addition, we confirmed the expression of genes involved in NAD^+^ synthesis and axon death signaling that are involved in NAD^+^ metabolism by measuring respective mRNA abundance in fly heads (Figure 2 —figure supplement 1B). We note that the expression and activity of NAD^+^ metabolic enzymes can be readily detected fly heads.”

Figure 2 —figure supplement 2:

“Prompted by such a significant change in the NAD^+^ metabolic flux, we wondered whether the change could alter the expression of genes involved in NAD^+^ metabolism or axon death signaling. However, besides the expected significant increase of the Gal4-mediated expression of NMN-D and NMN-D^dead^, we did not observe any notable changes (Figure 2 —figure supplement 2). Our observations demonstrate that the expression of NMN-D alone is sufficient to change the NAD^+^ metabolic flux*,* thereby significantly lowering NMN levels without affecting NAD^+^ in *Drosophila* heads; they serve as an excellent tissue for metabolic analyses.”

Figure 4 —figure supplement 1: see above.

Targeted protein profiling:

We also measured protein levels of an endogenously GFP-tagged dNmnat in different genetic backgrounds (e.g., *Actin–Gal4,* and when NMN-D, NMN-D^dead^, or mNAMPT is expressed). Both short and long isoforms (PC and PA, respectively) can be detected with the knock-in of GFP in the endogenous locus of *dnmnat* (Li-Kroeger et al., 2018). We could not detect any significant changes in dNmnat protein levels, as indicated in Author response image 1 (left, Western blot; right, quantification of GFP levels of PA and PC, respectively).

We feel that it is important to show this data here in the rebuttal. But we feel that it would add confusion to the readers in the manuscript.

**Author response image 1. sa2fig1:** 

The use of murine NMNAT (mNAMPT):We decided to use mouse NAMPT (mNAMPT) because it was readily available by Giuseppe Orsomando (Amici et al., 2017), and because we did not have access to human NAMPT (hNAMPT).

2) The authors introduce a novel/new PncC antibody however there are no details on this, its generation, availability, or how it was tested. This should be included in the manuscript. The antibody detects with several bands. The authors speculate that this could be a degradation product but nothing substantial is shown – can the authors support this claim more?

Why mNAMPT:

We decided to use mouse NAMPT (mNAMPT) because it was readily available by Giuseppe Orsomando (Amici et al., 2017), and because we did not have access to human NAMPT (hNAMPT).

We agree with the observation that under physiological conditions, the expression of mNAMPT does not change NMN. However, we argue that after injury, once dNmnat is degraded, the additional NMN synthesis provided by mNAMPT expression (in addition to dNrk), leads to a faster NMN accumulation. It is supported by the observation that NMNAT2 is more labile than NAMPT in mammals (Gilley and Coleman, 2010; Stefano et al., 2015).

3) Olfactory receptor neuron degeneration assays are shown in Figure 1(D and E) but no data is presented alongside this to back it up. The data for these images should be presented as well.

See answer to point 2 of reviewer #1 above.

Reviewer #3 (Recommendations for the authors):Overall, the manuscript describes a large set of reagents for the study of NAD metabolism in injury-induced degeneration. In my opinion, the conclusions regarding NMN are highly interesting because the effects are so strong. In fact, I think the authors could characterize this in a bit more detail. For example, the NAD concentration in axons should still decline after injury because NMNAT is degraded. Why are the axons not dying? Does NMN also affect NMNAT stability? Or is the NAD present in the axons before injury so stable it can provide for the lifetime?

Does NMN affect NMNAT stability:

See response to reviewer #2 (Pete Williams) on page 7, under “targeted protein profiling”

NAD stability:

We cannot measure NAD^+^ stability in severed preserved axons in the fly head. It would be interesting to assess the stability of the metabolite, but we feel that this question would exceed the scope of this manuscript.

The authors could also combine their metabolome (which is actually not very well explained in the manuscript text) and genetic analyses and test a number of predictions from the pathway diagrams as well. For example, does the NaDS pathway become essential in NMN-D overexpressing flies?

Explanation of metabolomic analyses:

We felt that the explanation, e.g., how we measured the actual metabolites, is well documented in Materials and Methods.

Does the NaDS pathway become essential in NMN-D overexpressing flies:

The reviewer raised an interesting and important question that we could readily address. We tested whether NaDS (now renamed to Nadsyn) is required for NMN-D-mediated protection. In neurons expressing NMN-D, we targeted Nadsyn by RNAi. After injury, the protection afforded by NMN-D is partially reverted, arguing that NMN activates dSarm, while NAD^+^ prevents dSarm activation (Figure 5 —figure supplement 2).

In the Results section, we changed a new paragraph:

The preservation of severed axons provided by NMN-D is partially reverted by RNAi-mediated knockdown of Nadsyn

“We have now established that NMN activates dSarm to trigger the degeneration of severed axons in *Drosophila*. While NMN induces a conformational change in a pocket of the ARM domain, NAD^+^ prevents this activation by competing for the same pocket (Bratkowski et al., 2020; Figley et al., 2021; Jiang et al., 2020; Zhao et al., 2019). We therefore wanted to test whether the preservation provided by lower NMN is reverted by a simultaneous reduction of NAD^+^ synthesis. We generated NMN-D-expressing neurons containing RNAimediated knockdown of Nadsyn (*nadsyn^RNAi^*). At 7 dpa, the 100% preservation provided by NMN-D was partially reduced to 60% by *nadsyn^RNAi^* in vivo (Figure 5 —figure supplement 2). This observation supports the degenerative NMN and the protective NAD^+^ function by activating and inhibiting dSarm in injury-induced axon degeneration in *Drosophila.”*

In the Discussion section, we also added a new paragraph:

“While NMN activates dSarm by inducing a conformational change in a pocket of the inhibitory ARM domain, NAD^+^ competes for the same pocket, acting as an inhibitor of dSarm activation (Jiang et al., 2020). Our simultaneous manipulation of NMN and NAD^+^ levels (by NMN-D expression and *nadsyn^RNAi^,* respectively) further supports that this competition is crucial in *Drosophila* to regulate dSarm activity and, consequentially, axon degeneration after axotomy in vivo*.”*

Writing – could be more confident/straightforward. Examples– "It remains currently unclear whether NMNd expression levels determine the extent of preservation." Does not sound like the best justification for an extensive study– "Our observations highlight that the combination of Cas9 and sgRNAs has to be carefully determined in each tissue targeted by CRISPR/Cas9." Every fly researcher knows this by now…– The introduction makes it sound as if NMNAT2 degradation is induced by injury. Is that the case? I thought it was constantly degraded and needs to be replenished.– Are axed and hiw explained properly in the introduction?

We carefully re-read and adjusted our writing to make it more concise. While we agree that “Every fly researcher knows this by now…”, we nevertheless felt that it is important to state our observations.

We also changed the introduction, to clarify the confusion of NMNAT2:

“In mammals, SARM1 activation is tightly controlled by metabolites in the NAD^+^ biosynthetic pathway. The labile enzyme nicotinamide mononucleotide adenylyltransferase 2, NMNAT2 is constantly transported into the axon, where it is degraded by the E3 ubiquitin ligase PAM-Highwire-Rpm-1 (PHR1) and mitogen-activated protein kinase (MAPK) signaling (Babetto et al., 2013; Gilley and Coleman, 2010; Walker et al., 2017). This steady state results in sufficient NMNAT2 that consumes nicotinamide mononucleotide (NMN) to synthesize NAD^+^. Upon axonal injury, axonal transport halts. Subsequently, NMNAT2 is rapidly degraded.

Explanation of *axed* and *hiw* in the introduction: We cover the essentials of Axed and Hiw in the introduction. Especially in the paragraph where we highlight conserved mechanisms in *Drosophila,* and notable differences.

References:

Amici, A., Grolla, A.A., Grosso, E.D., Bellini, R., Bianchi, M., Travelli, C., Garavaglia, S., Sorci, L., Raffaelli, N., Ruggieri, S., et al. (2017). Synthesis and Degradation of Adenosine 5′-Tetraphosphate by Nicotinamide and Nicotinate Phosphoribosyltransferases. Cell Chem Biol *24*, 553-564.e4.

Gilley, J., and Coleman, M.P. (2010). Endogenous Nmnat2 is an essential survival factor for maintenance of healthy axons. PLoS Biol *8*, e1000300.

Li-Kroeger, D., Kanca, O., Lee, P.-T., Cowan, S., Lee, M.T., Jaiswal, M., Salazar, J.L., He, Y., Zuo, Z., and Bellen, H.J. (2018). An expanded toolkit for gene tagging based on MiMIC and scarless CRISPR tagging in *Drosophila*. *ELife 7*.

MacDonald, J.M., Beach, M.G., Porpiglia, E., Sheehan, A.E., Watts, R.J., and Freeman, M.R. (2006). The *Drosophila* cell corpse engulfment receptor Draper mediates glial clearance of severed axons. Neuron *50*, 869– 881.

Stefano, M.D., Nascimento-Ferreira, I., Orsomando, G., Mori, V., Gilley, J., Brown, R., Janeckova, L., Vargas, M.E., Worrell, L.A., Loreto, A., et al. (2015). A rise in NAD precursor nicotinamide mononucleotide (NMN) after injury promotes axon degeneration. Cell Death Differ.